# Importance of El Niño reproducibility for reconstructing historical $CO_2$ flux variations in the equatorial Pacific

Michio Watanabe[1], Hiroaki Tatebe[1], Hiroshi Koyama[1], Tomohiro Hajima[1], Masahiro Watanabe[2], and
Michio Kawamiya[1]

[1]Research Institute for Global Change, Japan Agency for Marine-Earth Science and Technology (JAMSTEC), 3173-25, Showa-machi, Kanazawa-ku, Yokohama, Kanagawa, 236-0001, Japan.
[2]Atmosphere and Ocean Research Institute, the University of Tokyo, 5-1-5, Kashiwanoha, Kashiwa, Chiba, 277-8564, Japan.

*Correspondence to:* Michio Watanabe (michiow@jamstec.go.jp)

**Abstract.** Based on a set of climate simulations utilizing two kinds of Earth System Models (ESMs) to which observed ocean hydrographic data are assimilated with an exactly same data assimilation procedure, we have clarified that successful simulation of observed air–sea $CO_2$ flux variations in the equatorial Pacific is tightly linked with the reproducibility of physical air–sea coupled processes. When an ESM with weaker amplitude of ENSO (El Niño Southern Oscillations) than observations was used for historical simulations with the ocean data assimilation, observed equatorial anticorrelated relationship between the sea surface temperature (SST) and air–sea $CO_2$ flux on interannual-to-decadal timescales cannot be represented. The simulated $CO_2$ flux anomalies were upward (downward) during El Niño (La Niña) periods in the equatorial Pacific. The reason is that nonnegligible correction term on the governing equation of ocean temperature, which was added through the ocean data assimilation procedure, caused anomalously spurious equatorial upwelling (downwelling) during El Niño (La Niña) periods, which brought more (less) subsurface layer water rich in dissolved inorganic carbon (DIC) to the surface layer. On the other hand, in the historical simulations where the observational data were assimilated into the other ESM with more realistic ENSO representation, the correction term associated with the assimilation procedure was kept small enough so as not to disturb an anomalous advection-diffusion balance for the equatorial ocean temperature. Consequently, spurious vertical transport of DIC and resultant positively-correlated SST and air–sea $CO_2$ flux variations did not occur. Thus, the reproducibility of the tropical air–sea $CO_2$ flux variability with data assimilation can be significantly attributed to the reproducibility of ENSO in an ESM. Our results suggest that, when using data assimilation to initialize ESMs for carbon cycle predictions, the reproducibility of the internal climate variations in the model itself is of great importance.

## 1 Introduction

Since the industrial revolution, vast quantities of greenhouse gases (e.g., $CO_2$) have been released into the atmosphere through human activities such as fossil fuel use and land use change. Increased atmospheric $CO_2$ concentration leads to global warming, while both the oceanic and the terrestrial ecosystems absorb atmospheric $CO_2$ and are considered to

work to relax the progress of the global warming (Sabine et al., 2004; Doney et al., 2009a, 2014; Le Quéré et al., 2009, 2010, 2016). Observation-based studies have reached consensus that significant interannual variability of the air–sea $CO_2$ flux (hereafter, CO2F) exists in some specific regions such as the equatorial Pacific and high latitudes of both hemispheres (e.g., Park et al., 2010; Valsala and Maksyutov, 2010; Landschützer et al., 2014; Rödenbeck et al., 2014), and the variation of
CO2F associated with El Niño–Southern Oscillation (ENSO) in the equatorial Pacific has been highlighted in many previous observation-based and simulation-based studies (Keeling and Revelle, 1985; Feely et al., 1997, 1999; Jones et al., 2001; Obata and Kitamura, 2003; McKinley et al., 2004; Patra et al., 2005). When El Niño event occurs in the equatorial Pacific, dissolved inorganic carbon (DIC) concentration in the surface layer decreases due to lesser supply of the cold DIC-rich subsurface water to the surface layer than normal years because of weaker equatorial upwelling associated with weaker trade
winds (Le Borgne et al., 2002; Feely et al., 2004; Doney et al., 2009a, 2009b). Correspondingly, CO2F anomaly is downward during El Niño, and vice-versa during La Niña. Le Borgne et al. (2002) estimated that upwelling of DIC-rich subsurface water accounts for up to 70% of CO2F variation in the equatorial Pacific, while the other 30% is attributable to the variation of wind speed and biological processes. Accordingly, to estimate and predict variations of $CO_2$ uptake by the global ocean on timescales of several years, it would be informative to consider first the variations in the equatorial Pacific
associated with ENSO.

The Paris Agreement is an agreement within the United Nations Framework Convention on Climate Change (UNFCCC, 2015) providing the framework of measures from 2021 to 2030 to act against climate change. The goal of the Paris Agreement is to restrict the rise of the global mean surface air temperature to well below 2 °C relative to the preindustrial level. If greenhouse gas emissions continue to increase at their current rate, Earth's surface will warm by 1.5 °C
within ~20 years relative to the preindustrial state as reported in the fifth assessment report of the Intergovernmental Panel on Climate Change (IPCC, 2013). In this context, comprehensive understanding of the changes in the carbon cycle over previous years is essential for accurate predictions of the global carbon cycle, including natural variations, which will assist in evaluation of future $CO_2$ emission reductions (Kawamiya et al., 2020).

For future climate predictions, data assimilation procedures are incorporated into climate models in order to
synchronize simulated climatic states in the model with observations, that is, initialization of climate models. By incorporating data assimilation procedures into Earth System Models (ESMs), it will be possible to reproduce and predict variations in biogeochemical properties (Brasseur et al., 2009; Tommasi et al., 2017a, 2017b; Park et al., 2018). This includes an assessment of the predictability of CO2F on decadal timescale for the global ocean (Li et al., 2016, 2019).

Focusing on CO2F fluctuations associated with ENSO in the equatorial Pacific, Dong et al. (2016) analyzed the
results of the Earth system models (ESMs) that participated in the Coupled Model Intercomparison Project (CMIP) Phase 5 (CMIP5; Taylor et al., 2012), which contributed to the Fifth Assessment Report (AR5) of the Intergovernmental Panel on Climate Change (IPCC, 2013). They showed that only some ESMs could reproduce the observed anticorrelated relationship between SST and CO2F. This suggests that our understanding of ENSO and associated global carbon cycle variations are

still insufficient. For reliable prediction of future $CO_2$ uptake on interannual-to-decadal timescales, it is necessary to understand physical air–sea coupled process and associated carbon cycle variations in the equatorial Pacific.

In this study, utilizing two kinds of ESMs to which observed ocean hydrographic data are assimilated, we attempted to identify the key processes to reproduce the observed historical air–sea $CO_2$ flux variations in the equatorial Pacific. The remainder of this paper is organized as follows. Sect. 2 provides a brief description of the models used in this study, and the derived results are presented in Sect. 3. Finally, a short discussion and a summary are presented in Sect. 4.

## 2 Methods

### 2.1 Model Description

In this study, we have conducted four experiments, NEW-assim, NEW, OLD-assim, and OLD. In NEW-assim and NEW, we used the MIROC-ES2L (Hajima et al., 2020) and in OLD-assim and OLD, we used the MIROC-ESM (Watanabe, S. et al., 2011). The former is newly developed for CMIP Phase 6 (CMIP6; Eyring et al., 2016), while the latter is an official model of CMIP5. The physical core model of MIROC-ES2L is MIROC5.2, which is a minor update of MIROC5 (Watanabe, M. et al. 2010; Tatebe et al. 2018), while that of MIROC-ESM is MIROC3m (K-1 model developers, 2004). The horizontal resolution of the atmospheric component of MIROC-ES2L (MIROC-ESM) has T42 spectral truncation (i.e., approximately 300 km) with 40 (80) vertical levels up to 3 hPa (0.003 hPa). The oceanic component of MIROC-ES2L has a horizontal tripolar coordinate system. In the spherical coordinate portion south of 63°N, the longitudinal grid spacing is 1°, while the meridional grid spacing varies from approximately 0.5° near the equator to 1° in mid-latitude regions. There are 62 vertical levels in a hybrid σ–z coordinate system, the lowermost of which is located at the depth of 6300 m. The oceanic component of MIROC-ESM has a horizontal bipolar coordinate system: the longitudinal grid spacing of the oceanic component is approximately 1.4°, while the latitudinal grid intervals vary gradually from 0.5° at the equator to 1.7° near both poles. There are 44 vertical levels in a hybrid σ–z coordinate system, the lowermost of which is located at the depth of 5300 m. The resolutions in MIROC-ES2L are higher than in MIROC-ESM. In particular, 31 (21) of the 62 (44) vertical layers in MIROC-ES2L (MIROC-ESM) are within the upper 500 m of depth. The increased number of vertical layers in MIROC-ES2L has been adopted in order to better represent the equatorial thermocline.

In NEW-assim and OLD-assim, we used the ESMs that incorporated the same simple scheme for ocean data assimilation, which comprised an incremental analysis update (IAU; Bloom et al., 1996; Huang et al., 2002). This technique is relatively simple compared to more elaborate ones such as ensemble Kalman filter and four-dimensional variational method, but is widely used for decadal climate predictions (e.g., Mochizuki et al., 2010; Tatebe et al., 2012). Positive aspects of IAU is relatively low computational cost, which enables decadal-to-centennial scale integration and a variety of parameter sensitivity experiments. In the IAU, during the analysis interval from $t = 0$ to $t = \tau$, the governing equation including a

correction term for temperature and salinity ($X$) is written as follows:

$$\frac{dX}{dt} = \text{adv.} + \text{diff.} + F + \frac{\alpha}{\tau} \Delta X^a, \qquad (1)$$

where adv. is the advection term, diff. is the diffusion term, $F$ is the surface flux term, and the final term on the right-hand side is the correction term with $\alpha$ as a constant, and $\Delta X^a$ as the analysis increment. We employed the values of $\tau = 1$ day and $\alpha = 0.025$ and the IAU was applied at depths between the sea surface and 3000 m (Tatebe et al., 2012). The analysis increment is calculated from $\Delta X^a = X^a(0) - X(0)$, where $X^a(0)$ is the analysis and $X(0)$ is the model first guess at $t = 0$; this term is kept unchanged during the analysis interval from $t = 0$ to $t = \tau$. For $X^a(0)$, we used observed anomalies with respect to observed monthly mean climatology during 1961–2000. For $X(0)$, simulated anomalies in NEW-assim (OLD-assim) with respect to monthly mean climatology in NEW (OLD) were used. Such a scheme often called 'anomaly assimilation' or 'anomaly initialization' is also used in many previous studies (e.g., Smith et al., 2007; Keenlyside et al., 2008; Pohlmann et al., 2009; Li et al., 2016, 2019; Sospedra-Alfonso and Boer, 2020). The monthly objective analysis of ocean temperature and salinity (Ishii and Kimoto, 2009) are assimilated into the model as $X^a$, with linear interpolation to daily data. Because observed DIC concentration are sparse in space and time, only ocean hydrographic data are used for data assimilation in the present study. Also, any atmospheric observations/reanalysis are not applied.

Both of NEW and OLD are the exactly same as the historical simulations designated by CMIP6 and CMIP5 protocols, respectively, with three ensemble members for each which are bifurcated from arbitrary years of the corresponding preindustrial control simulations. The ocean data assimilation experiments, NEW-assim and OLD-assim, are bifurcated from NEW and OLD at the year 1946, respectively, and they are integrated up to the year 2005. Note that the data assimilation experiments are driven with the same external forcings as in the historical simulations. In the later sections, the model results in 1961–2005 are analyzed.

## 2.2 Estimating pCO₂ change at the sea surface

CO2F depends on the difference in $CO_2$ partial pressure between the sea and the air, i.e.:

$$\text{CO2F} = K(\text{pCO}_2 - \text{pCO}_2^{\text{air}})(1 - \gamma), \qquad (2)$$

where $\text{pCO}_2$ ($\text{pCO}_2^{\text{air}}$) is the $CO_2$ partial pressure in the sea (air), $\gamma$ is the fraction of sea ice, and $K = k\alpha$ is the $CO_2$ gas transfer coefficient, where $k$ represents the $CO_2$ gas transfer velocity (Wanninkhof, 1992, 2014) and $\alpha$ represents the solubility of $CO_2$ in seawater (Weiss, 1974). The $CO_2$ gas transfer velocity $k$ is a function of wind speed and the Schmidt number (Wanninkhof, 1992). This study investigated the reproducibility of the anticorrelated relationship between CO2F and SST and therefore the direction of the flux is important. As $K$ does not affect the direction and the flux variation due to ENSO has larger amplitude in terms of $\text{pCO}_2$ than $\text{pCO}_2^{\text{air}}$ (Dong et al., 2017), the direction of flux is governed by the variation in $\text{pCO}_2$. Consequently, we evaluated the $\text{pCO}_2$ change at the sea surface in the equatorial Pacific.

Seawater pCO$_2$ values depend on temperature (T), salinity (S), DIC concentration, and total alkalinity (Alk); therefore, the change of pCO$_2$ can be expanded as follows:

$$\Delta pCO_2 = C(T) + C(S) + C(DIC) + C(Alk) + Res., \qquad (3)$$

where C(X) = $(\partial pCO_2/\partial X)\Delta X$ (X=T, S, DIC, Alk) is the pCO$_2$ change due to the change in X (X=T, S, DIC, Alk), and Res., which includes second-order terms (Takahashi et al., 1993), was estimated so that the left-hand side and right-hand sides in Eq. (3) are equal in this study. In Sect. 3, we evaluate the CO2F and pCO$_2$ variations in the equatorial Pacific in NEW-assim, NEW, OLD-assim, and OLD, and we calculate each term in Eq. (3) for each experiment.

## 2.3 Observation and reanalysis dataset

To assess CO2F, ocean temperature, and wind speed of the model output, we used observation or reanalysis datasets. As the CO2F dataset, we used the SOM-FFN (Landschützer et al., 2016, 2017, 2018). It is an estimate based on the ocean surface CO$_2$ observation data collection, SOCATv3 (Bakker et al., 2016), and provides monthly data since 1982. It shows significant interannual variation of CO2F in some specific regions such as the equatorial Pacific and high latitudes of both hemispheres (Figure S1). In Sect. 3, we focus on the CO2F in the Niño3 region (5°S–5°N, 150°W–90°W) which shows notable variation of CO2F in the equatorial Pacific. This region is also the maximum variability region for SST (Gill, 1980). As the SST dataset, the observational COBE-SST2 (Ishii et al., 2005; Hirahara et al., 2014) was used. The JRA-55 reanalysis (Kobayashi et al., 2015) was used  for wind speed dataset.

145

## 3 Results

### 3.1 CO$_2$ flux and pCO$_2$ anomalies in Niño3 region

Horizontal maps for correlation coefficients between simulated and observed CO2F are shown in Figure 1. The model output data were ensemble mean and linearly interpolated into the SOM-FFN grid. Note that the data were not detrended, and one-year running mean filter is applied to monthly COF2 anomalies in 1982–2005 before calculating the correlation coefficients in accordance with the period for which the SOM-FFN dataset is available. CO2F in NEW-assim shows a positive correlation with SOM-FFN in the equatorial Pacific region (Figure 1a) where significant interannual variations of CO2F are found (Figure S1). On the other hand, CO2F in OLD-assim (Figure 1c) is negatively correlated in the equatorial Pacific. The timeseries in the Niño3 region of both one-year running mean SST (hereafter, NINO3-SST) and CO2F (hereafter, NINO3-CO2F) anomalies simulated with NEW-assim (OLD-assim) are shown in Figure 1b (Figure 1d). Here, the data were detrended and monthly anomalies were calculated with respect to the 1971–2000 monthly mean climatology. The correlation coefficients between NINO3-SST and NINO3-CO2F anomalies in NEW-assim, OLD-assim,

and observation are –0.50, 0.44, and –0.75, respectively (Table 1). The results in NEW-assim are consistent with the observations, while those in OLD-assim are not. The correlation coefficients between NINO3-SST and NINO3-CO2F anomalies in NEW and OLD are –0.85 and –0.67, respectively (Table 1 and Figure S2). Note that OLD could capture the observed anticorrelated relationship between NINO3-SST and NINO3-CO2F anomalies, but OLD-assim could not reproduce this relationship.

As the vertical direction of CO2F is determined mainly by pCO$_2$ at the sea surface (see Eq. (2)), we further estimated each term in Eq. (3) for each model output (Figure 2). $\partial pCO_2/\partial X$ in C(X) term in Eq. (3) (X=T, S, DIC, or Alk) was estimated based on the climatological annual mean T, S, DIC, and Alk at the sea surface within the Niño3 region in each experiment. $\Delta X$ in C(X) ($\Delta pCO_2$ on the left-hand side of Eq. (3)) is the variation of X (pCO$_2$) associated with ENSO and was calculated by averaging monthly mean X (pCO$_2$) anomalies regressed on NINO3-SST anomalies over the entire Niño3 region. Note that the NINO3-SST anomalies are standardized by standard deviation. In the following, we describe the anomalies during El Niño periods, while the opposite applies during La Niña periods. In NEW-assim, NEW, and OLD, pCO$_2$ decreases because the effect of the decrease in pCO$_2$ with decreasing DIC concentrations is larger than that of the increase in pCO$_2$ with warming (Figure 2). In OLD-assim, however, the effect of the increase in pCO$_2$ with warming is larger than that of OLD, and the decrease in pCO$_2$ with decreasing DIC concentrations is smaller than that of OLD, resulting in an increase in pCO$_2$. As noted in Sect. 1, previous studies (Le Borgne et al., 2002; Feely et al., 2004; Doney et al., 2009a, 2009b) showed that variability in upwelling during ENSO events dominates the equatorial Pacific CO2F variations through its regulation of DIC. In the following, we discuss the temperature and vertical velocity changes associated with ENSO along the Equator.

## 3.2 DIC and vertical velocity changes

A cross section of the monthly ocean temperature anomalies regressed onto the standardized monthly mean NINO3-SST anomalies along the equatorial Pacific is presented in Figures 3 and S3, together with the climatological annual mean depths of the 18, 20, and 22 °C isotherms. Here, monthly temperature anomalies were calculated with respect to the 1971–2000 monthly mean climatology. The observational temperature anomalies as well as the climatological isotherms are derived from the monthly objective analysis of ocean temperature (Ishii and Kimoto, 2009). Amplitudes of the positive (negative) equatorial temperature anomalies in the upper (lower) layer of the eastern (western) equatorial Pacific in NEW are larger than in OLD and are closer to observations. The intensity of ENSO defined as the standard deviation of detrended one-year running mean NINO3-SST anomalies from 1961 to 2005 is shown in Table 2. The intensity of ENSO in NEW is estimated to be 1.17 °C (Table 2), a bit stronger than the observation (0.80 °C). On the other hand, the intensity of ENSO in OLD is 0.43 °C, which is about a half as large as that in observations. In addition, the climatological mean thermocline in NEW is tighter than in OLD and is closer to observations. The improvement in ENSO reproducibility in NEW is attributable mainly to two updates in the model configuration. The first is implementation of an updated plume model for cumulus

convection with multiple cloud types where lateral entrainment rate varies vertically depending on the surrounding environment (Chikira and Sugiyama, 2010). The state-dependent lateral entrainment affects the strength of convectively-induced air–sea coupled processes in the eastern tropical Pacific, and thus the ENSO amplitude in the model. More details are described in Watanabe et al. (2010). The second is reduction of numerical diffusion by introducing highly-accurate tracer advection scheme in the ocean and by increasing vertical resolutions (Prather, 1986). The equatorial thermocline in the climatic-mean state of the tropical Pacific is more diffuse in OLD than in observation, which is partly arisen from numerical diffusion, especially in vertical advection (Tatebe and Hasumi, 2010), and this model bias is much alleviated in NEW. Correspondingly, so-called thermocline mode (e.g., Imada et al., 2006) becomes more effective and ENSO amplitude becomes larger in NEW. As the ENSO amplitude in NEW is larger than in OLD, the variation of the equatorial trade wind, which causes anomalous equatorial vertical velocity, is also larger in NEW.

To assess the variations of zonal wind associated with ENSO, we estimated the 10 m zonal wind anomalies over the NINO4 region (5°S–5°N, 160°E–150°W; the dotted line boxes in Figure 1a) which are regressed onto the NINO3-SST anomalies (Table 3). Niño4 region is the maximum variability region for the equatorial trade wind (Figure S4). Hereafter, the above-mentioned regression coefficient is referred to as wind feedback. The positive value of wind feedback in NEW (0.92 m $s^{-1}$ $K^{-1}$) indicates an westerly wind anomalies during El Niño, and this is consistent with that evaluated from the observational dataset, i.e., 1.02 m $s^{-1}$ $K^{-1}$. The wind feedback in OLD (0.46 m $s^{-1}$ $K^{-1}$) is about half of NEW and the observation.

Cross sections of the monthly upward water velocity and DIC concentration anomalies along the equator regressed onto the standardized NINO3-SST anomalies in NEW (OLD) are shown in Figure 4a and 4c (Figure 4b and 4d), respectively. By reproducing wind feedback that is consistent with the observation, the westerly wind anomalies during El Niño periods in NEW (Figure S4c) is comparable to that of the JRA-55 reanalysis (Figure S4i), leading to weakening of upward vertical velocity of approximately $5 \times 10^{-6}$ m $s^{-1}$ (Figure 4a). This weakening of upward vertical velocity causes decrease in surface DIC in the eastern equatorial Pacific during El Niño periods (Figure 4c). In OLD, the smaller wind feedback and associated smaller westerly wind anomalies than in the JRA-55 reanalysis (Figure S4g) leads to weakening of upward vertical velocity of just $10^{-6}$ m $s^{-1}$ in the equatorial Pacific (Figure 4b). Although the ENSO signal in OLD is weaker than the observation, because of decrease in upward vertical velocity from normal years, the surface DIC concentration decreases during El Niño periods (Figure 4d). This is consistent with Dong et al. (2016), showing that OLD is able to qualitatively reproduce the negative correlation between SST and DIC concentration anomalies in the eastern equatorial Pacific (Figure S2b).

Next, we examined the correction term in temperature due to the data assimilation, i.e., temperature analysis increment, the final term on the right-hand side of Eq. (1), and the variations in vertical velocity and DIC concentration. Anomalies of monthly mean temperature analysis increments, vertical velocity, and DIC concentration along the equator regressed onto the standardized NINO3-SST anomalies are shown in Figure 5. The maximum absolute value of the equatorial temperature analysis increment in NEW-assim is found at 10–40 m depths in the eastern equatorial Pacific,

shallower than the depth of the thermocline (Figure 5a). In NEW-assim, the wind feedback is 0.92 m s$^{-1}$ K$^{-1}$ (Table 3), which is of the same magnitude to that in NEW (0.92 m s$^{-1}$ K$^{-1}$), and the surface wind anomalies still shows similar pattern to that of the NEW (Figure S4a–d). The westerly wind anomalies in NEW-assim leads to weakening of upward vertical velocity along the equator during El Niño periods (Figure 5c). To assess the variation in equatorial vertical velocity associated with ENSO, we estimated the anomalies of the vertical velocity at the depth of the 20 °C isotherm (the depth of the thermocline) in the Niño3 region which are regressed onto the NINO3-SST anomalies. Hereafter, the regression coefficient is referred to as vertical velocity feedback. The vertical velocity feedback in NEW-assim is estimated to be –4.5 × 10$^{-7}$ m s$^{-1}$ K$^{-1}$, which is not significantly different from that in NEW (–3.9 × 10$^{-7}$ m s$^{-1}$ K$^{-1}$) (Table 3). The negative value of vertical velocity feedback in NEW-assim indicates the weakening of upward vertical velocity at the depth of the thermocline during El Niño periods in the eastern equatorial Pacific (Figure 5c). The weakening of upward vertical velocity causes lesser supply of the DIC-rich subsurface water to the surface layer, leading to the decrease in surface DIC concentration (Figure 5e). In OLD, the temperature variations associated with ENSO at the depth of the thermocline in the eastern equatorial Pacific is smaller than observed (see Figure 3b and 3c), so that the correction term forces to raise the equatorial water temperature by 0.16 × 10$^{-6}$ °C s$^{-1}$ during El Niño periods in order to realize observed temperature variations (Figures 5b and S3b). The wind feedback in OLD-assim is 0.48 m s$^{-1}$ K$^{-1}$ (Table 3), which is the same as in OLD, and the map of the wind speed anomalies shows a similar pattern to that of the OLD (Figure S4e–h); however, the warming due to data assimilation procedure during El Niño periods reduces density, leading to low-pressure anomalies. This results in anomalous cyclonic circulation and convergence, and thus enhancement of upward vertical velocity at the depth of the thermocline (Figure 5d). The vertical velocity feedback in OLD-assim is 4.1 × 10$^{-7}$ m s$^{-1}$ K$^{-1}$, which has an opposite sign to OLD, –4.9 × 10$^{-7}$ m s$^{-1}$ K$^{-1}$ (Table 3). The positive value of vertical velocity feedback indicates the enhancement of upward vertical velocity at the depth of the thermocline during El Niño periods, which is inconsistent with observations. This spurious enhancement of upward vertical velocity during El Niño periods causes the increase in the surface DIC concentration (Figure 5f), leading to positive correlation between SST and CO2F (Figure 1d), contrary to observations. We have to note here that even in the NEW-assim, the vertical velocity distribution (Figure 5c) is still different from NEW (Figure 4a) because of the temperature analysis increment. As already discussed, the intensity of ENSO in NEW is slightly stronger than observed (Table 2). In addition, the period of ENSO, which is defined as the peak of the power spectrum of one-year running mean NINO-SST, is 5.0 years in NEW, which is longer than 3.5 years of observations (see Table 2). Because the ENSO characteristics in NEW are not perfectly consistent with observations, model nature, namely responses of vertical velocity and DIC concentration in ENSO, are still distorted by the temperature analysis increment even in NEW-assim. This indicates that further model improvements are needed.

**4 Discussion and Summary**

In the present study, comparing the results of two ESMs to which observed ocean hydrographic data are assimilated, we have clarified that representation of the processes in the equatorial climate system is important to reproduce the observed anticorrelated relationship between SST and CO2F in the equatorial Pacific. In the case where the ocean temperature and salinity observations were assimilated into an ESM with weaker amplitude of ENSO than observations, the correction term on the governing equation of the ocean temperature, which was introduced in the data assimilation procedure, caused spurious upwelling (downwelling) anomalies along the equator during El Niño (La Niña) periods, leading to more (less) supply of the DIC-rich subsurface water to the surface layer. Due to the resultant increase (decrease) of the surface DIC concentration, the upward (downward) CO2F anomalies during El Niño (La Niña) periods was induced, which was inconsistent with observation. In the case where the ocean temperature and salinity observations were assimilated into the other ESM with rather realistic ENSO representation, anticorrelated relationship between SST and CO2F was reproduced.

Focusing on the CO2F fluctuations associated with ENSO in the equatorial Pacific, Dong et al. (2016) analyzed the results of the CMIP5 ESMs. They showed that only a portion of CMIP5 ESMs (including MIROC-ESM) could reproduce the observed anticorrelated relationship between SST and CO2F. Bellenger et al. (2014) evaluated the reproducibility of ENSO in the CMIP5 models. They reported that most CMIP5 climate models and ESMs underestimate the amplitude of the wind stress feedback by 20%–50%, and that only 20% of CMIP5 models have relative error within 25% of the observed value. There are many ESMs where the ENSO characteristics and/or the SST-CO2F relationship are inconsistent with observations. Causes of this discrepancy should be addresses in future studies through, for example, multi-model analysis, and also process-based uncertainty estimation will be further required in initialized climate and carbon predictions as well as projections by ESMs.

**Data availability**

The model outputs of MIROC-ES2L (Hajima et al., 2019) are available through the Earth System Grid Federation (ESGF) (https://doi.org/10.22033/ESGF/CMIP6.5602). The model outputs of MIROC-ESM is also available at ESGF, https://esgf-node.llnl.gov/projects/cmip5/. The CMIP6 forcing data is version 6.2.1, and the CMIP5 forcing data is described at https://pcmdi.llnl.gov/mips/cmip5/forcing.html. The SOM-FFN dataset is available at https://www.ncei.noaa.gov/access/ocean-carbon-data-system/oceans/SPCO2_1982_2015_ETH_SOM_FFN.html. The JRA-55 reanalysis wind dataset is available at https://jra.kishou.go.jp/JRA-55/index_en.html. The COBE-SST2 dataset is available at https://www.esrl.noaa.gov/psd/data/gridded/data.cobe2.html. The postprocessing scripts used for this research and the data used in the figures can be obtained online (https://osf.io/mpk52).

## Author contribution

MiW, HT, MaW, and MK contributed to the experiment design. MiW and HK embedded the ocean data assimilation system into the ESMs. MiW and TH performed the experimental simulations. MiW analyzed the model output and drafted the paper. All authors discussed the results, and commented on and edited the manuscript.


## Competing interests

The authors declare that they have no conflict of interest.

## Acknowledgments

This work was supported by the Integrated Research Program for Advanced Climate Models (TOUGOU) Grant Numbers JPMXD0717935457 and JPMXD0717935715 from the Ministry of Education, Culture, Sports, Science and Technology, MEXT, Japan. We thank B. Barton, J. Jardine, M. Payo Payo, C. Unsworth and one anonymous reader for helpful comments. We also thank three anonymous reviewers provided many helpful suggestions for improving the article.

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

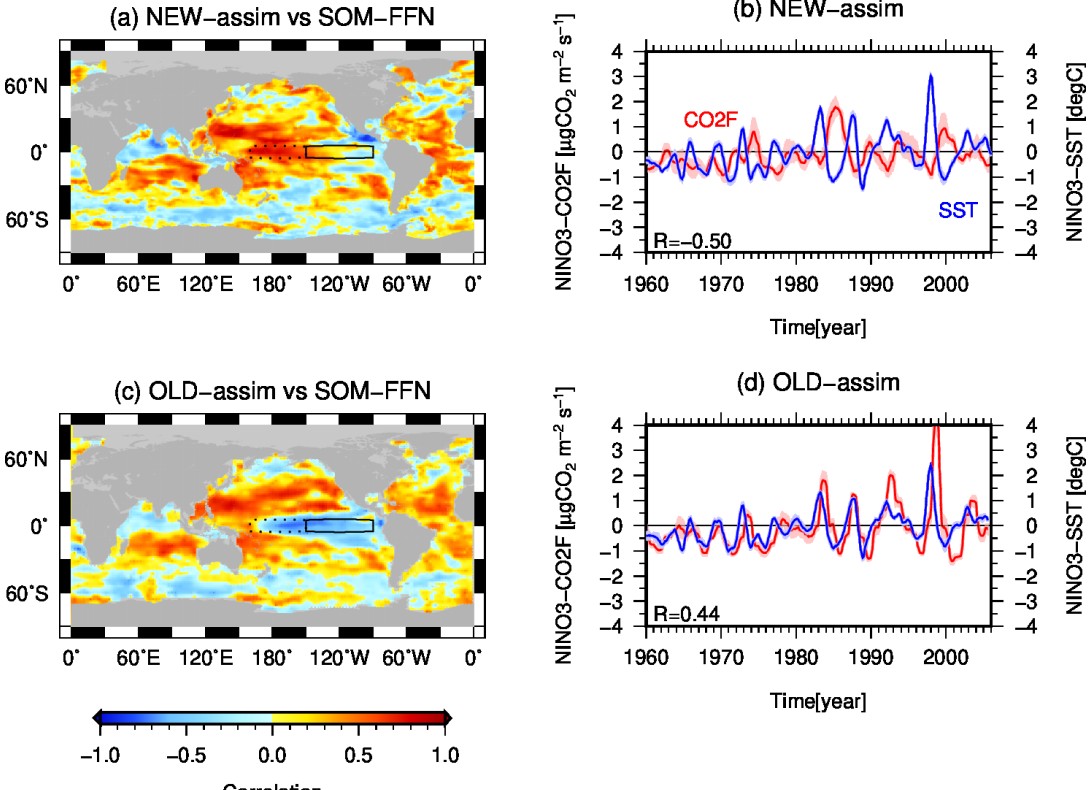

Figure 1. (a, c) Maps for correlation coefficient between monthly CO2F anomalies derived from SOM-FFN and that of (a) NEW-assim and (c) OLD-assim. The analysis period is from 1982 to 2005. The solid line boxes show Niño3 region (5°S–5°N, 90°–150°W) and the dotted line boxes show Niño4 region (5°S–5°N, 160°E–150°W). (b, d) Timeseries of the detrended NINO3-SST (blue line) and NINO3-CO2F (red line, positive upward) anomalies simulated with (b) NEW-assim and (d) OLD-assim. Values plotted are the one-year running mean, and shading in (b) and (d) shows the ensemble spread (1σ). R denotes the correlation coefficients between the detrended ensemble mean NINO3-SST and NINO3-CO2F anomalies, with one-year running mean filter applied.

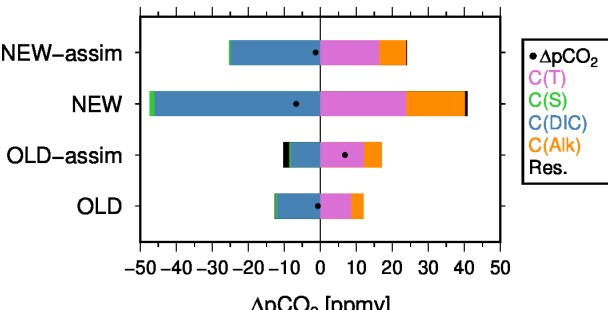

Figure 2. Changes in $pCO_2$ regressed onto standardized NINO3-SST anomalies ($\Delta pCO_2$) (dots) and its decomposition with changes in X (X=T, S, DIC, Alk), C(X), as well as Res. (Eq. (3)) evaluated in NEW-assim, NEW, OLD-assim, and OLD. See text for details in calculation of $\Delta pCO_2$, C(X), and Res.

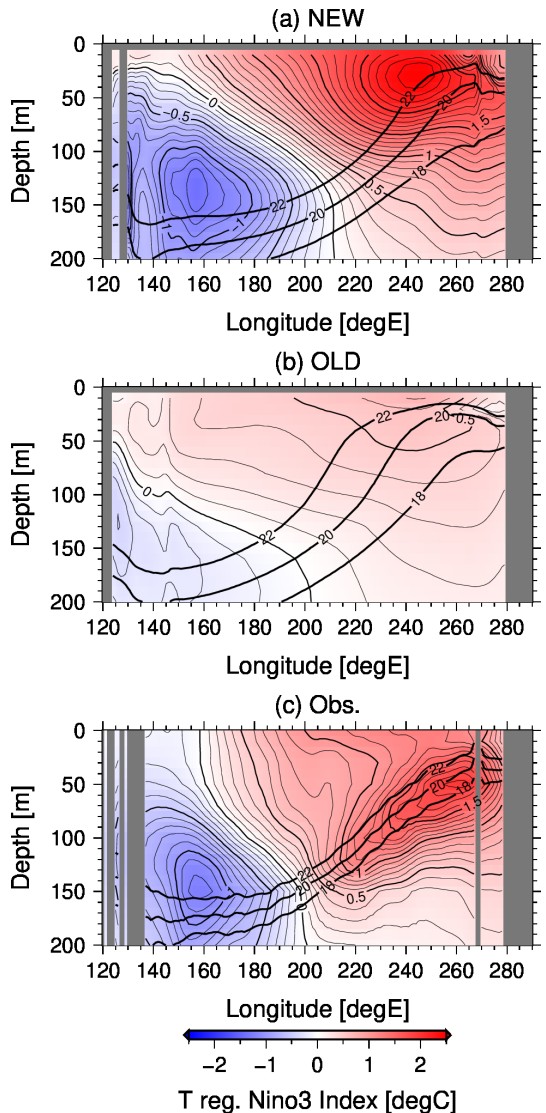

**Figure 3. Anomalies of equatorial ocean temperature regressed onto standardized NINO3-SST anomalies for NEW (top), OLD (middle), and observations (bottom). Contour interval is 0.1 °C. Thick solid lines indicates the climatological-mean isotherms of the 18, 20, and 22 °C.**

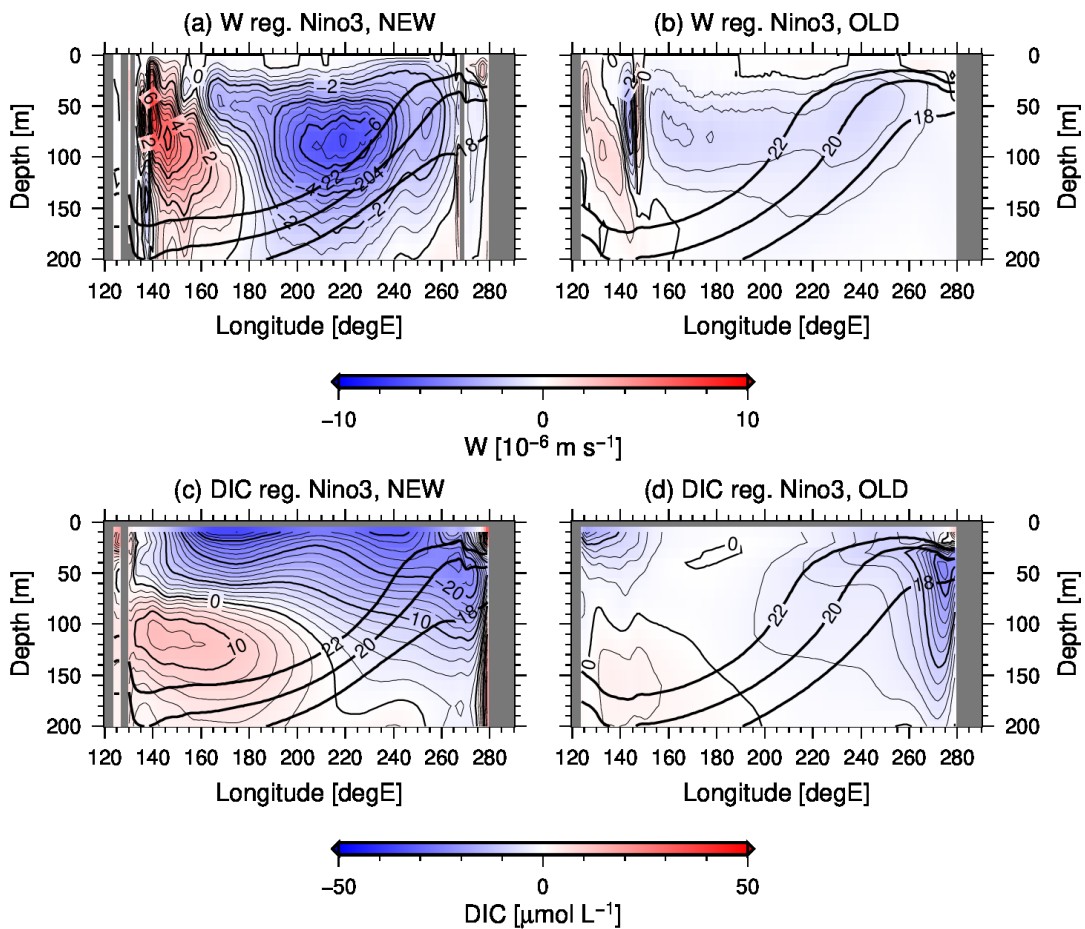

**Figure 4.** Anomalies of equatorial vertical velocity (upper panels) and DIC (lower panels) regressed onto standardized NINO3-SST anomalies for NEW (left) and OLD (right). Contour intervals are $0.5 \times 10^{-6}$ m s$^{-1}$ in (a,b) and 2 μmol L$^{-1}$ in (c,d), respectively. Thick solid lines indicates the climatological-mean isotherms of the 18, 20, and 22 °C.

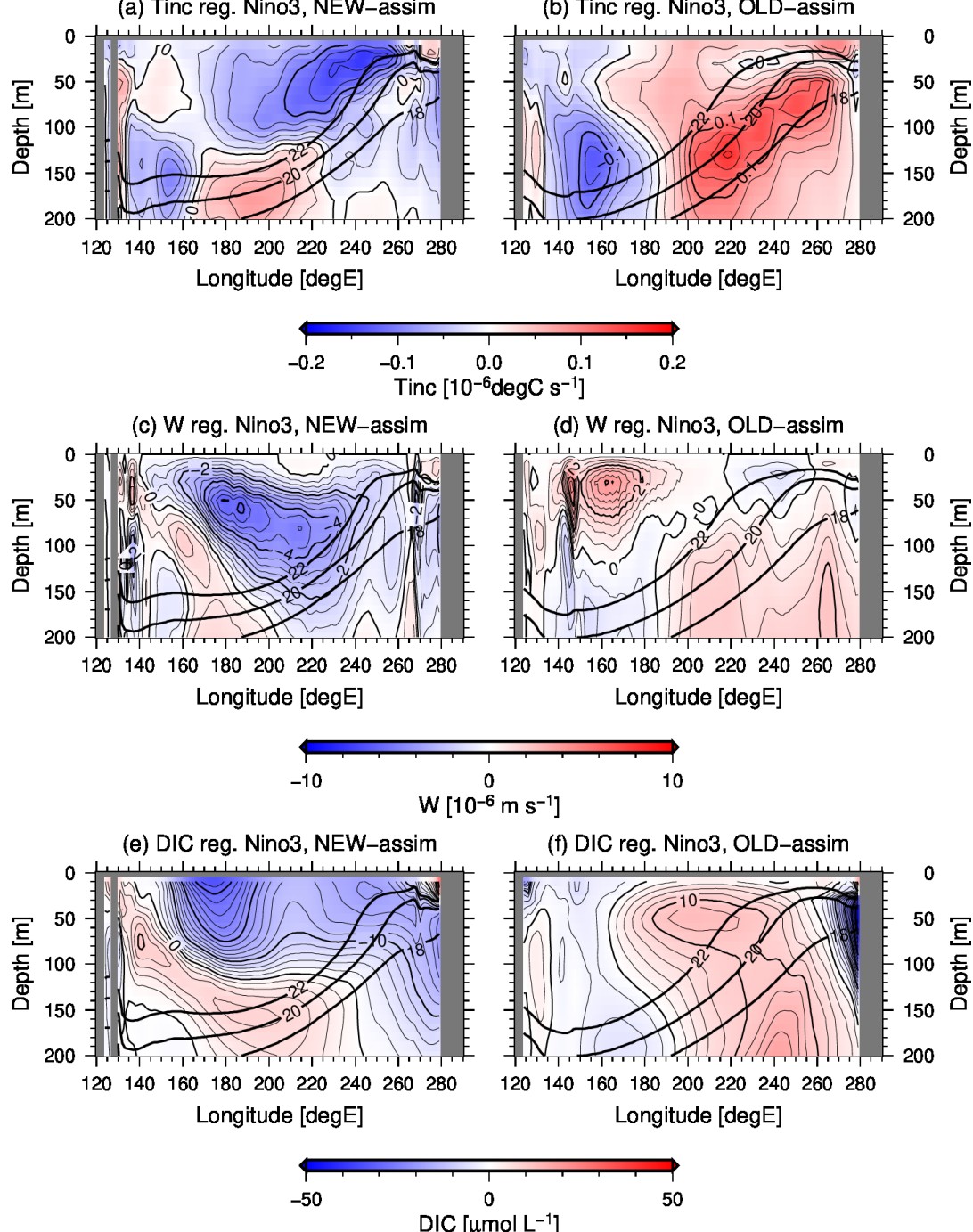

**Figure 5.** Equatorial temperature analysis increments (top panels), vertical velocity anomalies (middle panels) and DIC anomalies (bottom panels) regressed onto standardized NINO3-SST anomalies for NEW-assim (left) and OLD-assim (right). Contour intervals are $0.02 \times 10^{-6}$ °C s$^{-1}$ in (a,b), $0.5 \times 10^{-6}$ m s$^{-1}$ in (c,d) and 2 µmol L$^{-1}$ in (e,f), respectively. Thick solid lines indicates the climatological-mean isotherms of the 18, 20, and 22 °C.

**Table 1. Correlation coefficients between detrended one-year running mean NINO3-SST and NINO3-CO2F anomalies in NEW-assim, NEW, OLD-assim, OLD, and observations. The correlations coefficients in NEW-assim, NEW, OLD-assim, and OLD are for the period from 1961 to 2005 (Figures 1 and S2), and that in observations are for the period from 1982 to 2005.**

|              | NEW-assim | NEW   | OLD-assim | OLD   | Observation |
|--------------|-----------|-------|-----------|-------|-------------|
| Corr. Coeff. | –0.50     | –0.85 | 0.44      | –0.67 | –0.75       |

**Table 2. The intensity and period of ENSO in NEW, OLD, and observations calculated from the one-year running mean NINO3-SST anomalies for the period from 1961 to 2005.**

|                      | NEW  | OLD  | Observation |
|----------------------|------|------|-------------|
| Intensity of ENSO [℃] | 1.17 | 0.43 | 0.80        |
| Period of ENSO [yr]  | 5.0  | 4.5  | 3.5         |

**Table 3. The wind feedback and the vertical velocity feedback in NEW-assim, NEW, OLD, and OLD-assim. The wind feedback is computed as the monthly 10 m zonal wind anomalies in the Niño4 region regressed onto the monthly NINO3-SST anomalies, and the vertical velocity feedback the monthly vertical velocity anomalies at the depth of the 20 °C isotherm in the Niño3 region regressed onto the monthly NINO3-SST anomalies. The wind feedback is also evaluated from the observation dataset.**

|                                          | NEW-assim            | NEW                  | OLD-assim           | OLD                  | Observation |
|------------------------------------------|----------------------|----------------------|---------------------|----------------------|-------------|
| Wind feedback [m s$^{-1}$ K$^{-1}$]      | 0.92                 | 0.92                 | 0.48                | 0.46                 | 1.02        |
| Vertical velocity feedback [m s$^{-1}$ K$^{-1}$] | $-4.5\times10^{-7}$ | $-3.9\times10^{-7}$ | $4.1\times10^{-7}$ | $-4.9\times10^{-7}$ | N/A         |