# Peer review of "Importance of El Niño reproducibility for reconstructing historical CO2 flux variations in the equatorial Pacific"

_Ocean Science, 2020_

## Referee Comment (RC1) · Anonymous Referee #1 · 19 May 2020

The authors present an interesting work to compare the relationship of air-sea carbon fluxes to ENSO in the equatorial Pacific simulated by the two earth system models with assimilation and without assimilation, which are developed by the same institute. What's more interesting in this paper is that the old earth system model with assimilation generated an incorrect upwelling during the El Nino period, which led to a great problem in the simulation of carbon fluxes, but the new earth system model did not. Although this work has not made much contribution to the study of the response mechanism of carbon fluxes to ENSO in the equatorial Pacific, it will be very helpful to the people who are interested in assimilation or the model development, especially to those who are interested in the simulation of the carbon cycle process in the equatorial Pacific, if we can find out why the old and new models have different performances after the same assimilation method is added. The content of the article fits within the scope of OS, but much work needs to be done before publication to refine the theme of the article, highlight key points, and give a more detailed discussion on the conclusions. Major points: 1 Abstractïij ŽGreat changes are needed to reduce the description of the study significance and increase the discussion of the final result. 2 Some descriptions need to be supplemented, such as the vertical range of assimilation. Are the temperature and salinity at the bottom of the mixed layer assimilated? That another content needs to be added is to compare the differences in the simulation of ENSO between the two models with assimilation and without assimilation, such as the periodicity and amplitude of ENSO. 3 In the old model with and without assimilation, the response of 10-m wind speed over the sea surface to NINO3- SST does not change significantly, but the response of sea water vertical velocity to NINO3- SST changes greatly (Figure 4). Dose the meridional wind change significantly?

4 Line 196, "howerver, the strong heating causes upwelling of DIC rich waters in the subsurface layers (Figure 6b)." Why does this strong heating occur? Is the simulated value of sea water temperature in the old model during the El Nino period lower than the data used for the assimilation? Please discuss in detail the reasons for the abnormal upwelling during the El Nino period in the old model with assimilation.

5 After assimilation is added to the new earth system model, the response of upwelling anomalies to NINO3- SST is weakened (comparison of Fig. 6 with Fig. 8). This change in the response is actually similar to that in the old model. Does this mean that the current assimilation method is not suitable to the earth system model?

Minor points:

1 Line 119, "three ensemble members". How were the ensemble experiments conducted? Were the initial fields of these experiments different?

2 The statement of Line 149-151 is error. $(\partial pCO2/\partial T)\Delta T$ is not the term of changing

the solubility of CO2.

3 How was the "temperature increment" calculated?

4 Line 236-238, "The correlation between SST and CO2F in the equatorial Pacific is consistently represented only in the case where the ocean temperature and salinity observations are assimilated into NEW." This statement is ambiguous, because both OLD and NEW experiments can produce the relationship between the SST and CO2F.

5 Overall, the manuscript needs to be improved, including some language errors.

---

## Referee Comment (RC2) · Anonymous Referee #2 · 21 May 2020

This paper describes the benefits of advanced data assimilation method in advanced CMIP6-class climate model compared to CMIP5 model. The model results and their mechanisms have been well described in this manuscript. I would recommend this paper is acceptable in this Ocean Science Journal with some support analysis based on comparison using observations to verify the assimilation skills, which could be much elevating the values of this paper.

L52. Can we discard the biological pump on the results, especially in the La Nina states? Author represented NINO3-CO2F correlation coefficients, which means both El Nino and La Nina events. As we know, decreasing the phytoplankton in El Nino

event could affect the CO2F variability modulated by DIC solely but I wonder whether the strong positive bloom in La Nina event could absorb the CO2 into the ocean. If then, the better performance of the phytoplankton assimilation skill can be a key to elevate the better CO2F skill.Composite analysis between CO2F at El Nino and La Nina and taking difference of them to see the asymmetry would elevate the biological influence on CO2F in this model. If then, you may provide supporting figures of chlorophyll skills in this model using satellite-derived chlorophyll concentration using such as ESA-CCI (https://esa-oceancolour-cci.org) or GlobalColour in Hermes (http://hermes.acri.fr).

L142.   What about observational skills in the region for CO2F associated with ENSO compared to NEW-assim skill -0.41?   This can be depending on the definitions of regional and temporal scales but as you cited Dong et al (2016) represents above 0.6 skills in many CMIP5-class model (it seems like opposite sign for CO2F). Of course they do not have assimilation but do you think the ENSO-CO2F skill is generated by some limitations coming from assimilation? Otherwise you may add comparison between OLD and NEW model correlation (or regression) skill of ENSO-CO2F without assimilation (freerun) to argue this issue as a table likewise arranging skills of OLD, OLD-assim, NEW, NEW-assim and with skill of available SST reanalysis and psudo observation data of CO2 flux at least single observation dataset such as using Landschutzer et al 2016 (link: https://www.nodc.noaa.gov/ocads/oceans/SPCO2_1982_2015_ETH_SOM_FFN.html ), opened to public or data-based estimates of carbon cycle variability ( http://www.bgc-jena.mpg.de/CarboScope/?ID=oc ), which is needed by personal contact to access. If then, you may add some figures and discussions in chapter 3.1 for comparison of ENSO-related CO2F skills between in observation, OLD, and NEW model in spatial and temporal scales.  If the results are significant, this could be providing the most benefit in this paper and persuading rest of results being reasonable.  According to this, you may see some figures and references in Hongmei Li et al. 2019 as you cited.

---

## Referee Comment (RC3) · Anonymous Referee #3 · 25 May 2020

This study is an important contribution for understanding ENSO and carbon fluxes variations in the equatorial Pacific. The authors have investigated the processes in regulating the relationship between ENSO and carbon fluxes in assimilations with nudging ocean temperature and salinity based on two MIROC models, i.e., OLD MIROC-ESM and NEW MIROC-ES2L. They demonstrated that the ability of model in producing correct amplitude of ENSO is crucial for reproduction of the air-sea CO2 flux variations in coherence with ENSO. Both the storyline and the writing are clear. However, there are still some unclear aspects listed as below, I would expect the authors further clarify them and improve the manuscript.

[Figure]

1. It is exciting to see the NEW model shows promising results of the anticorrelation between ENSO and air-sea $CO_2$ flux, which the OLD model couldn't capture well especially the magnitude of ENSO. As revealed by Dong et al. (2016), most CMIP5 models could not capture the relationship right. It would be helpful to have some discussion on which key model developments do improve the representation of ENSO magnitude in the NEW model? A paragraph of discussion on this will provide advices for other modeling centers.

2. ENSO is an air-sea coupled system, it involves both ocean and atmosphere processes. In this study, both OLD-assim and NEW-assim only nudge ocean temperature and salinity, the atmosphere ran freely without any data nudging. I have couple of questions here: i) Does the IAU apply to every ocean level including the ocean surface? ii) How is the atmosphere part for instance winds treated? As the ocean part has strong nudging, the atmosphere should be adjusted accordingly, the mismatch of ocean and atmosphere would cause some spurious circulation. iii) Why is this spurious upwelling only found in the OLD-assim? iv) Is the spurious upwelling obvious in the climatological mean state in OLD-assim comparing with the OLD? A comparison of climatology in the nudged data and the model free runs will help understand this point. v) Would a different assimilation method, e.g., including atmospheric circulation nudging, end up with a different conclusion?

3. Line 32: ". . .warm by 1.5C within ∼20 years. . ." -> ". . .warm by 1.5C within ∼20 years relative to the preindustrial state"

4. Line 87: "This remainder.." -> "The remainder. . ."

5. Combining Fig. 5 and Fig. 7, Fig. 6 and Fig. 8 will help readers for the comparison of OLD and NEW.

6. Line 234-236: "In this research, the same simple data assimilation scheme is incorporated into two ESMs, OLD in which the ENSO amplitude is about half the observed value and NEW with improved reproducibility of ENSO." Is this statement of ENSO amplitude based on the free runs of the two models? It would be helpful to add panels of ENSO amplitude in the free runs with OLD and NEW models in Fig. 1.

7. Line 237: "…is consistently represented…" here needs to be rephrased to make it clearer, e.g., the anticorrelation relationship between SST and CO2F.

───────────────────────

---

## Author Response (AR1)

Dear Dr. Hoppema,

Thank you very much for handling our manuscript. Following the reviewer's comment, we have revised the manuscript. Below, the reviewers' comments are indicated in black text and our responses are indicated in red text. Line numbers below refer to the tracked-changes version of the manuscript, although in the author comments in the interactive discussion, line numbers refer to the revised, non-tracked-changes version manuscript. In addition to the comments from the reviewers, we received some comments from researchers of the National Oceanographic Centre, UK., personally. We have revised the manuscript taking into account their comments as well. Their comments and our replies are also included below.

**Reply to referee #1:**

Thank you very much for invaluable comments and suggestions on our original manuscript. According to the comments, we have revised the manuscript. In addition, following the comment we received personally from the National Oceanographic Centre, UK. (For their comments and our reply, see Pages 15–24), we have combined Section 3.2 and 3.3 in the original manuscript and reorganized. We think the revised manuscript is now more readable. We hope that the revised manuscript meets your approval and will be more suitable for publication in the journal.

**Reply to comments:**

(Referee #1) "The authors present an interesting work to compare the relationship of air-sea carbon fluxes to ENSO in the equatorial Pacific simulated by the two earth system models with assimilation and without assimilation, which are developed by the same institute. What's more interesting in this paper is that the old earth system model with assimilation generated an incorrect upwelling during the El Nino period, which led to a great problem in the simulation of carbon fluxes, but the new earth system model did not. Although this work has not made much contribution to the study of the response mechanism of carbon fluxes to ENSO in the equatorial Pacific, it will be very helpful to the people who are interested in assimilation or the model development, especially to those who are interested in the simulation of the carbon cycle process in the equatorial Pacific, if we can find out why the old and new models have different performances after the same assimilation method is added.

The content of the article fits within the scope of OS, but much work needs to be done before publication to refine the theme of the article, highlight key points, and give a more detailed discussion on the conclusions."

Thank you for taking the time to review the manuscript. We have revised the manuscript in accordance with the following comments. We will answer them point by point.

Major points:
1 Abstract Great changes are needed to reduce the description of the study significance and increase the discussion of the final result.

We have completely rewritten the abstract. First, the first two sentences in the abstract in the original manuscript have been removed for conciseness and clearness on the purpose of this study. We then have described in more detail what led to the failure of reproducing the observed anticorrelation between SST and CO2F in the eastern equatorial Pacific with the ESM with smaller-than-observed amplitude of ENSO.

2 Some descriptions need to be supplemented, such as the vertical range of assimilation. Are the temperature and salinity at the bottom of the mixed layer assimilated? That another content needs to be added is to compare the differences in the simulation of ENSO between the two models with assimilation and without assimilation, such as the periodicity and amplitude of ENSO.

In this study, the observed anomalies are assimilated into the ocean models at depths between the sea surface and 3000 m. We have rewritten Lines 257–258 of the revised manuscript as follows:
"In addition, the IAU was applied at depths between the sea surface and 3000 m, with the values of $\tau = 1$ day and $\alpha = 0.025$ (Tatebe et al., 2012)."
The original manuscript did not clearly state that we used "anomaly assimilation". We have added the following sentences in Lines 254–257 in the revised manuscript:
"For $X^a(0)$ and $X(0)$, we used anomalies from monthly mean climatology during 1961–2000 in observations and models, respectively. Such a scheme often called 'anomaly assimilation' or 'anomaly initialization' is used in many previous studies (e.g., Smith et al., 2007; Keenlyside et al., 2008; Pohlmann et al., 2009; Li et al., 2016, 2019; Sospedra-Alfonso and Boer, 2020)."
We have added Table 2 showing intensities and periods of ENSO for NEW, OLD, and

observations in the revised manuscript. The discussion on the intensities (periods) of ENSO has been added in Lines 373–375 (612–616) in the revised manuscript. The ENSO intensities and periods for NEW-assim and OLD-assim are the same as the observed values because of data assimilation.

3 In the old model with and without assimilation, the response of 10-m wind speed over the sea surface to NINO3-SST does not change significantly, but the response of sea water vertical velocity to NINO3-SST changes greatly (Figure 4). Dose the meridional wind change significantly?

In OLD-assim, the warming due to data assimilation procedure during El Niño periods reduces density at the depth of the thermocline in the eastern equatorial Pacific, leading to enhancement of upward vertical velocity. The patterns of zonal and meridional wind speed variation in OLD and OLD-assim are similar. To describe the mechanism causing vertical velocity anomalies more clearly, we have added Figure S3 as a supplement and modified Lines 601–605 in the revised manuscript as follows:

"The wind feedback in OLD-assim is 0.48 m s$^{-1}$ K$^{-1}$ (Table 3), which is the same as in OLD, and the map of the wind speed anomalies shows a similar pattern to that of the OLD (Figure S3e–h); however, the warming due to data assimilation procedure during El Niño periods reduces density, leading to low-pressure anomalies. This results in anomalous cyclonic circulation and convergence, and thus enhancement of upward vertical velocity at the depth of the thermocline (Figure 5d)"

4 Line 196, "howerver, the strong heating causes upwelling of DIC rich waters in the subsurface layers (Figure 6b)." Why does this strong heating occur? Is the simulated value of sea water temperature in the old model during the El Nino period lower than the data used for the assimilation? Please discuss in detail the reasons for the abnormal upwelling during the El Nino period in the old model with assimilation.

In this study, the observed anomalies are assimilated into the ocean models at depths between the sea surface and 3000 m (see Reply to Major points 2). In OLD, the temperature variations in the eastern equatorial Pacific is smaller than observed (Figure 3b and 3c), so that the correction term on the governing equation of the ocean temperature, which is introduced in the data assimilation procedure, forces to raise the equatorial water temperature during El Niño

periods in order to realize observed temperature variations. To clarify this process, we have modified Lines 598–601 in the revised manuscript as follows:

"In OLD, the temperature variations associated with ENSO at the depth of the thermocline in the eastern equatorial Pacific is smaller than observed (see Figure 3b and 3c), so that the correction term forces to raise the equatorial water temperature by $0.16 \times 10^{-6}\,°C\ s^{-1}$ during El Niño periods in order to realize observed temperature variations (Figure 5b). "

In order to describe the process in which warming causes the enhancement of upward vertical velocity more clearly, we have rewritten Lines 603–605 as follows:

"the warming due to data assimilation procedure during El Niño periods reduces density, leading to low-pressure anomalies. This results in anomalous cyclonic circulation and convergence, and thus enhancement of upward vertical velocity at the depth of the thermocline (Figure 5d)."

5 After assimilation is added to the new earth system model, the response of upwelling anomalies to NINO3- SST is weakened (comparison of Fig. 6 with Fig. 8). This change in the response is actually similar to that in the old model. Does this mean that the current assimilation method is not suitable to the earth system model?

This study points out that, before discussing the assimilation methods, the performance of the model itself needs to be improved. We think that the reproduction of the observed anticorrelated relationship between SST and CO2F in the equatorial Pacific in NEW-assim indicates the usefulness of the MIROC-ES2L and the data assimilation method we used in this study.

However, we have to admit that MIROC-ES2L and the data assimilation method we used is not perfect. As the referee #1 pointed out, the assimilation scheme modifies the distribution of vertical velocity. At 140W, the upward vertical velocity anomaly during El Niño periods was $-7 \times 10^{-6}$ m/s in New, but it changed to about $-5 \times 10^{-6}$ m/s in the NEW-assim. The change in the upward vertical velocity from NEW to NEW-assim may be due to the fact that the ENSO intensity is stronger and the period is longer than in the observations, and response of vertical velocity in ENSO is still distorted by the temperature analysis increment in NEW-assim. To point out that the response of vertical velocity in ENSO is still distorted by the temperature analysis increment in NEW-assim, we have added the following sentences in Lines 611–616:

"As already discussed, the intensity of ENSO in NEW is slightly stronger than observed (Table 2). In addition, the period of ENSO, which is defined as the peak of the power spectrum of one-year running mean NINO-SST, is 5.0 years in NEW, which is longer than 3.5 years of observations (see Table 2). Because the ENSO characteristics in NEW are somewhat

inconsistent with observations, model nature, namely responses of vertical velocity and DIC concentration in ENSO, are still distorted by the temperature analysis increment even in NEW-assim. This indicates that further model improvements are needed."

We think that the further development of ESM and the use of more advanced assimilation methods may improve the performance of the model. Further investigation is required to identify the best suitable method and why.

Minor points:

1 Line 119, "three ensemble members". How were the ensemble experiments conducted? Were the initial fields of these experiments different?

Both of NEW and OLD are the exactly same as the historical simulations designated by CMIP6 and CMIP5 protocols, respectively, and they have three ensemble members which are bifurcated from arbitrary years of the corresponding preindustrial control simulations. NEW-assim and OLD-assim are bifurcated from NEW and OLD at the year 1946, respectively. We have added the following sentences in Lines 262–265 in the revised manuscript:

"Both of NEW and OLD are the exactly same as the historical simulations designated by CMIP5 and CMIP6 protocols, respectively, and they have three ensemble members which are bifurcated from arbitrary years of the corresponding preindustrial control simulations. The ocean data assimilation experiments, NEW-assim and OLD-assim, are bifurcated from NEW and OLD at the year 1946, respectively, and they are integrated up to the year 2005."

2 The statement of Line 149-151 is error. $(\partial pCO2/\partial T)\Delta T$ is not the term of changing the solubility of CO2.

The phrase "in CO2 solubility" in Lines 149–151 in the original manuscript has been removed.

3 How was the "temperature increment" calculated?

The phrase "temperature increment" in the original manuscript has been changed to "temperature analysis increment" in the revised manuscript. The method for calculating the temperature analysis increment is described in Sect. 2.1 in the revised manuscript, but here is a brief introduction. The analysis increment during the analysis interval from $t = 0$ to $t = \tau$ is calculated from $\Delta X^a = X^a(0) - X(0)$, where $X^a(0)$ is the analysis and $X(0)$ is the model first guess

at $t = 0$; this term is held constant during the analysis interval $\tau = 1$ d. The monthly objective analysis data of ocean temperature and salinity (Ishii and Kimoto, 2009) were interpolated linearly to form daily analysis data, $X^a$.

4 Line 236-238, "The correlation between SST and CO2F in the equatorial Pacific is consistently represented only in the case where the ocean temperature and salinity observations are assimilated into NEW." This statement is ambiguous, because both OLD and NEW experiments can produce the relationship between the SST and CO2F.

We realized that the first paragraph of Discussion and Summary section, as in abstract (see Reply to Major points 1), should be a concise statement of what we found in the study. We have removed the description of the background from the first paragraph of Discussion and Summary in the original manuscript and rewritten it so that there was no ambiguity. The relevant sentence has been changed to:
"In the case where the ocean temperature and salinity observations were assimilated into the other ESM with rather realistic ENSO representation, anticorrelated relationship between SST and CO2F was reproduced." (Lines 652–653)

5 Overall, the manuscript needs to be improved, including some language errors. "

We have reviewed the entire manuscript and revised it in accordance with the comments. Thank you again for your comments.

**Reply to referee #2:**

Thank you very much for invaluable comments and suggestions on our original manuscript. Following the comments, we have revised the manuscript. In addition, following the comment we received personally from the National Oceanographic Centre, UK. (For their comments and our reply, see Pages 15–24), we have combined Section 3.2 and 3.3 in the original manuscript and reorganized. We think the revised manuscript is now more readable. We hope that the revised manuscript meets your approval and will be suitable for publication in the journal.

**Reply to comments:**

(Referee #2) "This paper describes the benefits of advanced data assimilation method in advanced CMIP6-class climate model compared to CMIP5 model. The model results and their mechanisms have been well described in this manuscript. I would recommend this paper is acceptable in this Ocean Science Journal with some support analysis based on comparison using observations to verify the assimilation skills, which could be much elevating the values of this paper.

Thank you very much for your comments.

L52. Can we discard the biological pump on the results, especially in the La Nina states? Author represented NINO3-CO2F correlation coefficients, which means both El Nino and La Nina events. As we know, decreasing the phytoplankton in El Nino event could affect the CO2F variability modulated by DIC solely but I wonder whether the strong positive bloom in La Nina event could absorb the CO2 into the ocean. If then, the better performance of the phytoplankton assimilation skill can be a key to elevate the better CO2F skill. Composite analysis between CO2F at El Nino and La Nina and taking difference of them to see the asymmetry would elevate the biological influence on CO2F in this model. If then, you may provide supporting figures of chlorophyll skills in this model using satellite-derived chlorophyll concentration using such as ESA-CCI (https://esa-oceancolour-cci.org) or GlobalColour in Hermes (http://hermes.acri.fr).

Thank you for your suggestion. We here examine the effect of the biological pump on $CO_2$ flux in the equatorial Pacific. First of all, we investigated whether NEW-assim captures the historical variations in the bloom magnitude associated with ENSO. Figure R1a shows the timeseries of simulated surface chlorophyll concentration anomalies averaged over the Niño3

region (hereafter NINO3-Chla) and NINO3-SST anomalies in NEW-assim. NINO3-Chla anomalies derived from the observational dataset Ocean Colour Climate Change Initiative (OC-CCI) dataset, Version 4.2, European Space Agency, is also shown. Here, monthly anomalies were calculated with respect to the 1998–2005 monthly mean climatology because OC-CCI dataset is only available since September 1997. The results of NEW-assim shows that NINO3-Chla increased during La Niña, and the correlation coefficient between NEW-assim and the observed values was estimated to be 0.60, indicating that NEW-assim is able to capture the variations in primary production associated with ENSO. It should be noted here that the variation of NEW-assim is larger than the variation of the observed values. Since NEW-assim captures the historical variations in the bloom magnitude associated with ENSO, we next calculate the average of NINO3-CO2F for El Niño, La Niña, and others (neutral), respectively (Figure R1b). Here, following Japan Meteorological Agency, "El Niño event" is defined as a phenomenon in which the five-month running mean of the NINO3-SST anomaly exceeds +0.5°C for six consecutive months or more, and "La Niña event" as a phenomenon in which the five-month running mean of the NINO3-SST anomaly is below −0.5°C for six consecutive months or more (Figure R1a). The anomaly of NINO3-CO2F averaged during El Niño periods is −0.43 $\mu gCO_2$ m$^{-2}$ s$^{-1}$, and that averaged during La Niña periods is 0.36 $\mu gCO_2$ m$^{-2}$ s$^{-1}$. The absolute value of NINO3-CO2F anomaly averaged during La Niña periods is 15% smaller than that of El Niño periods, which can be explained by the biological pumps during La Niña periods. However, the standard error bar of NINO3-CO2F during La Niña periods overlaps that during El Niño periods, so that the difference in NINO3-CO2F is not significant and we did not include these results in the revised manuscript. Further studies are needed to quantify the effect of biological pump.

[Figure]

Figure R1.    (a) Timeseries of the detrended NINO3-Chla for NEW-assim (orange line) and observations

(black). The blue line is the timeseries of the detrended NINO3-SST anomalies in NEW-assim. Values plotted are the one-year running mean and shading shows the ensemble spread (1σ). The El Niño and La Niña periods is indicated by light magenta and light cyan, respectively. (b) Absolute values of monthly mean NINO3-CO2F anomalies averaged over El Niño, La Niña, and other (neutral) periods, respectively, during the period from 1960 to 2005 simulated with NEW-assim. Error bars indicate the standard deviations of monthly mean NINO3-CO2F anomalies. Note that they are not the standard deviations of the absolute values of monthly mean NINO3-CO2F.

L142. What about observational skills in the region for CO2F associated with ENSO compared to NEW-assim skill -0.41? This can be depending on the definitions of regional and temporal scales but as you cited Dong et al (2016) represents above 0.6 skills in many CMIP5-class model (it seems like opposite sign for CO2F). Of course they do not have assimilation but do you think the ENSO-CO2F skill is generated by some limitations coming from assimilation? Otherwise you may add comparison between OLD and NEW model correlation (or regression) skill of ENSO-CO2F without assimilation (freerun) to argue this issue as a table likewise arranging skills of OLD, OLD-assim, NEW, NEW-assim and with skill of available SST reanalysis and psudo observation data of CO2 flux at least single observation dataset such as using Landschutzer et al 2016 (link: https://www.nodc.noaa.gov/ocads/oceans/SPCO2_1982_2015_ETH_SOM_FFN.html ), opened to public or data-based estimates of carbon cycle variability ( http://www.bgc-jena.mpg.de/CarboScope/?ID=oc ), which is needed by personal contact to access. If then, you may add some figures and discussions in chapter 3.1 for comparison of ENSO-related CO2F skills between in observation, OLD, and NEW model in spatial and temporal scales. If the results are significant, this could be providing the most benefit in this paper and persuading rest of results being reasonable. According to this, you may see some figures and references in Hongmei Li et al. 2019 as you cited."

Thank you for your suggestion. In order to discuss the correlation coefficients between CO2F and SST in each experiment, we have added Table 1 in the revised manuscript. We have recalculated the correlation coefficients between CO2F and SST in NEW-assim, and it was estimated to be –0.50. The absolute value of the correlation coefficient in NEW-assim is less than the absolute value of the correlation coefficient of NEW (–0.85). This is because model nature are somewhat distorted by the temperature analysis increment even in NEW-assim. In the revised manuscript, we have added the following sentence in Lines 614−616:
"Because the ENSO characteristics in NEW are not perfectly consistent with observations, model nature, namely responses of vertical velocity and DIC concentration in ENSO, are still

distorted by the temperature analysis increment even in NEW-assim. This indicates that further model improvements are needed."

To compare the maps for the correlation coefficients between CO2F from SOM-FFN and that from the NEW-assim or from OLD-assim, we have added the new figures in the revised manuscript (Figure 1a and 1c). CO2F in NEW-assim (Figure 1a) is positively correlated with SOM-FFN in the equatorial Pacific. On the other hand, CO2F in OLD-assim shows a negative correlation with SOM-FFN there. In Lines 320–326 in the revised manuscript, we have added the description of these figures. The description on SOM-FFN has been added in Sect. 2.3 in the revised manuscript.

**Reply to referee #3:**

Thank you very much for invaluable comments and suggestions on our original manuscript. We would like to answer the questions given by the referee and to describe how we have revised our manuscript point by point. In addition, following the comment we received personally from the National Oceanographic Centre, UK. (For their comments and our reply, see Pages 15–24), we have combined Section 3.2 and 3.3 in the original manuscript and reorganized. We think the revised manuscript is now more readable. We hope that the revised manuscript meets your approval and will be more suitable for publication in the journal.

**Reply to General comments:**

(Referee #3) "This study is an important contribution for understanding ENSO and carbon fluxes variations in the equatorial Pacific. The authors have investigated the processes in regulating the relationship between ENSO and carbon fluxes in assimilations with nudging ocean temperature and salinity based on two MIROC models, i.e., OLD MIROC-ESM and NEW MIROC-ES2L. They demonstrated that the ability of model in producing correct amplitude of ENSO is crucial for reproduction of the air-sea $CO_2$ flux variations in coherence with ENSO. Both the storyline and the writing are clear. However, there are still some unclear aspects listed as below, I would expect the authors further clarify them and improve the manuscript.

Thank you very much for your comments. We have reviewed the entire manuscript and revised it in accordance with the comments.

1. It is exciting to see the NEW model shows promising results of the anticorrelation between ENSO and air-sea $CO_2$ flux, which the OLD model couldn't capture well especially the magnitude of ENSO. As revealed by Dong et al. (2016), most CMIP5 models could not capture the relationship right. It would be helpful to have some discussion on which key model developments do improve the representation of ENSO magnitude in the NEW model? A paragraph of discussion on this will provide advices for other modeling centers.

The important model improvements in MIROC-ES2L was not stated in the original manuscript. We have added the description on it in Lines 376–454 in the revised manuscript. In brief, one is implementation of an updated plume model for cumulus convection with multiple cloud types

where lateral entrainment rate varies vertically depending on the surrounding environment. The other is reduction of numerical diffusion by introducing highly-accurate tracer advection scheme in the ocean and by increasing vertical resolutions.

2. ENSO is an air-sea coupled system, it involves both ocean and atmosphere processes. In this study, both OLD-assim and NEW-assim only nudge ocean temperature and salinity, the atmosphere ran freely without any data nudging. I have couple of questions here: i) Does the IAU apply to every ocean level including the ocean surface? ii) How is the atmosphere part for instance winds treated? As the ocean part has strong nudging, the atmosphere should be adjusted accordingly, the mismatch of ocean and atmosphere would cause some spurious circulation. iii) Why is this spurious upwelling only found in the OLD-assim? iv) Is the spurious upwelling obvious in the climatological mean state in OLD-assim comparing with the OLD? A comparison of climatology in the nudged data and the model free runs will help understand this point. v) Would a different assimilation method, e.g., including atmospheric circulation nudging, end up with a different conclusion?

i)    In this study, the observed temperature and salinity are assimilated into the ocean models at depths between the sea surface and 3000 m. To state this, we have rewritten Lines 257–258 in the revised manuscript as follows:
      "In addition, the IAU was applied at depths between the sea surface and 3000 m, with the values of $\tau = 1$ day and $\alpha = 0.025$ (Tatebe et al., 2012)."

ii)   In the atmosphere, data assimilation is not used. To clarify this, we have added the following sentence in Line 261 in the revised manuscript:
      "Also, any atmospheric observations/reanalysis are not applied."
      The ocean temperature and salinity observations were assimilated into ESMs and the atmosphere responds to them.

iii)  Here, we describe the anomalies during El Niño periods, while the opposite applies during La Niña periods. In OLD, the ENSO signal is weaker than the observation, so that the correction term on the governing equation of the ocean temperature forces to raise the equatorial water temperature in order to realize observed temperature variations. The warming due to data assimilation procedure reduces density, leading to enhancement of upward vertical velocity at the depth of thermocline. In NEW, on the other hand, because amplitudes of the equatorial temperature anomalies are larger than in OLD and are closer to observations, the correction term in NEW-assim arisen from

the assimilation procedure was kept small enough not to cause spurious enhancement of upward vertical velocity. To describe in more detail the mechanism by which upward vertical velocity in the equatorial Pacific in OLD-assim enhances during El Niño periods, we have rewritten Lines 598–605 in the revised manuscript as follows:

"In OLD, the temperature variations associated with ENSO at the depth of the thermocline in the eastern equatorial Pacific is smaller than observed (see Figure 3b and 3c), so that the correction term forces to raise the equatorial water temperature by $0.16 \times 10^{-6}$ °C s$^{-1}$ during El Niño periods in order to realize observed temperature variations (Figure 5b). The wind feedback in OLD-assim is 0.48 m s$^{-1}$ K$^{-1}$ (Table 3), which is the same as in OLD, and the map of the wind speed anomalies shows a similar pattern to that of the OLD (Figure S3e–h); however, the warming due to data assimilation procedure during El Niño periods reduces density, leading to low-pressure anomalies. This results in anomalous cyclonic circulation and convergence, and thus enhancement of upward vertical velocity at the depth of the thermocline (Figure 5d)."

iv) In this study, the observed temperature and salinity anomalies are assimilated into the ocean models at depths between the sea surface and 3000 m, which was not described in the original manuscript. Therefore, the climatological mean states of ocean temperature and salinity with assimilation are same with those without assimilation. In order to clarify that the observed anomalies are assimilated into the model in this study, we have added the following sentences in Lines 254–257:

"For $X^a(0)$ and $X(0)$, we used anomalies from monthly mean climatology during 1961–2000 in observations and models, respectively. Such a scheme often called 'anomaly assimilation' or 'anomaly initialization' is used in many previous studies (e.g., Smith et al., 2007; Keenlyside et al., 2008; Pohlmann et al., 2009; Li et al., 2016, 2019; Sospedra-Alfonso and Boer, 2020)."

v) Different assimilation techniques could make the model correlate better with the observations. Further investigation is required to identify the best suitable method and why. However, we think if the model itself does not perform well, the assimilation process leads to an unnatural circulation, as in OLD-assim in this study.

3. Line 32: "…warm by 1.5C within ~20 years…" -> "…warm by 1.5C within ~20 years relative to the preindustrial state"

Corrected. (Line 72 in the revised manuscript)

4. Line 87: "This remainder.." -> "The remainder. . ."

Corrected. (Line 134 in the revised manuscript)

5.Combining Fig. 5 and Fig. 7, Fig. 6 and Fig. 8 will help readers for the comparison of OLD and NEW.

Following the comment, Figures 5 and 7 (Figures 6 and 8) in the original manuscript have been combined into Figure 4 (Figure 5) in the revised manuscript.

6. Line 234-236: "In this research, the same simple data assimilation scheme is incorporated into two ESMs, OLD in which the ENSO amplitude is about half the observed value and NEW with improved reproducibility of ENSO." Is this statement of ENSO amplitude based on the free runs of the two models? It would be helpful to add panels of ENSO amplitude in the free runs with OLD and NEW models in Fig. 1.

We have added Table 2 in the revised manuscript, that shows the intensity and period of ENSO in NEW, OLD, and observations. We have also added Figure S2 showing the timeseries of the detrended NINO3-SST and NINO3-CO2F anomalies simulated by one ensemble member in OLD and NEW and that derived from the observation.

7. Line 237: ". . .is consistently represented. . ." here needs to be rephrased to make it clearer, e.g., the anticorrelation relationship between SST and CO2F. "

In order to state the results of this study more concisely and clearly in Discussion and Summary section, its first paragraph has been totally rewritten and the relevant sentence has been modified as follows:
"In the case where the ocean temperature and salinity observations were assimilated into the other ESM with rather realistic ENSO representation, anticorrelated relationship between SST and CO2F was reproduced." (Lines 652–653)

**Reply to comment from the National Oceanographic Centre, UK.:**

Thank you very much for invaluable comments and suggestions on our original manuscript. According to the comments, we have revised the manuscript. Here, we copied all your comments and answered to all your comments point by point using red font.

**Reply to comments:**

**Main comments:**

"Overall the study is important for the improvement of CMIP models and their ability to reproduce ENSO variability. The story is in good shape and we do not think new simulations are required. There are many points that need clarification and the wording needs tightening to avoid confusion in places. Some unanswered questions detailed below would improve this study and make it more widely applicable to other CMIP models. Overall, we think the study needs minor revisions.

Thank you for taking the time to review the manuscript. We have revised the manuscript in accordance with the following comments.

**Minor Comments:**

Abstract

Abstract needs to make it clearer what the research question - and answer is. Place the question clearly, perhaps phrase lines 13-15 with question. It is important to communicate in the abstract that the OLD model does not reproduce observations (it is assumed the reader already knows this), and the newer models does. The final line of the abstract is vague and could be (tersly) summarised with "new model is better than old". What are the consequences of this? Where does this work lead and what are the immediate implications?

Thank you for your comments. Following the comments, we have completely rewritten the abstract. To more concisely and clearly state the purpose of this study, the first two sentences in the original manuscript have been removed. We then have described in more detail what led to the failure of reproducing the observed anticorrelation between SST and CO2F with the ESM with smaller-than-observed amplitude of ENSO, and pointed out that the performance of the model is important when initializing an ESM.

Introduction

Introduction is long and takes a while to get to the main problem with the CMIP5 model. The main point of the paper, discrepancy between the observations and MIROC-ESM for El Niño amplitude and associated $CO_2$ flux, should be identified in the first paragraph more clearly rather than the end of paragraph 4 (lines 52-54).

Thank you for your advice. We have totally rewritten and shortened Introduction section in the revised manuscript. The main point of this manuscript described in fourth paragraph in the original manuscript has been moved to the first paragraph in the revised manuscript.

Lines 64-72, do we need the history of data assimilation in climate models to understand this paper?

We have removed the description of the history of data assimilation in climate models in the revised manuscript.

Lines 76-89, the last paragraph of the introduction should specify the question this paper will answer and set out the structure of the paper. The question is unclear. Instead, this paragraph has text about assimilation that should be in the methods section.

To clarify the purpose of this study in Introduction, we have moved some sentences describing ESMs and data assimilation methods in the original manuscript to Methods section in the revised manuscript.

Methods
Lines 99-104, the grid is very irregular was it interpolated? Is the model sensitive to grid (add ref)?

To the south of 63°N, spherical coordinates are used. Analyses of SST and $CO_2$ flux variations in the Niño3 region were performed using data from the original grid. In order to compare the air–sea CO2 flux of NEW-assim and OLD-assim to SOM-FFN dataset (Landschützer et al., 2016), model output is linearly interpolated into the SOM-FFN grid. To describe the interpolation, we have added the following sentence in Line 321 in the revised manuscript:
"The model output data were ensemble mean and linearly interpolated into the SOM-FFN grid."

Lines 99-104, the NEW model is deeper 5300 vs 6300. Is the model sensitive to this (add ref)? How is the model partitioned vertically? Sigma/z/hybrid coordinate? This is important since the stratification is a key part of your results.

The vertical level both in MIROC-ES2L and MIROC-ESM are in a hybrid σ–z coordinate system. We have added the phrase "in a hybrid σ–z coordinate system" in Line 149 and Line 152 in the revised manuscript, respectively. Since this study focuses on processes near the surface, we do not think the change in maximum depth has had much of an impact. Rather, increasing vertical resolution within the upper 500 m of depth has an impact. In order to describe the vertical resolution, we have added the following sentences in Lines 152–155 in the revised manuscript:

"The resolutions in MIROC-ES2L are higher than in MIROC-ESM. In particular, 31 (21) of the 62 (44) vertical layers in MIROC-ES2L (MIROC-ESM) are within the upper 500 m of depth. The increased number of vertical layers in MIROC-ES2L has been adopted in order to better represent the equatorial thermocline."

Line 118, the ensembles are only mentioned here. We need more detail. How are they different? Maybe use a table.

Both of NEW and OLD have three ensemble members which are bifurcated from arbitrary years of the corresponding preindustrial control simulations, and NEW-assim and OLD-assim are bifurcated from NEW and OLD, respectively. We have mentioned this by adding the following sentence in Lines 262–265 in the revised manuscript:

"Both of NEW and OLD are the exactly same as the historical simulations designated by CMIP6 and CMIP5 protocols, respectively, with three ensemble members for each which are bifurcated from arbitrary years of the corresponding preindustrial control simulations. The ocean data assimilation experiments, NEW-assim and OLD-assim, are bifurcated from NEW and OLD at the year 1946, respectively, and they are integrated up to the year 2005"

Line 133, equation is repeated in the text. Instead, define the variables here. For example, what is $(\partial pCO2/\partial Alk)\Delta Alk$?

In the revised manuscript, we have defined $C(X)= (\partial pCO_2/\partial X)\Delta X$ (X=T, S, DIC, Alk), as

pCO₂ change due to the change in X, and stated that Res. in Eq (3), which includes second-order terms (Takahashi et al., 1993), was estimated so that the left-hand side and right-hand sides in Eq. (3) are equal. (Lines 302–304 in the revised manuscript).

Methods is missing description of the boxes NINO3 and NINO4 for someone not familiar with ENSO analysis. Why are the boxes picked? A map would be useful here.

We analyzed CO2F in Niño3 region because this region shows maximum variability region for CO2F. We have added the map for standard deviations of CO2F anomalies derived from observation-based CO2F dataset SOM-FFN (Landschützer et al., 2016) (Figure S1). In the revised manuscript, to clarify why we focus on CO2F in this region, we have added the following sentences in Lines 310–313:

"It shows significant interannual variation of CO2F in some specific regions such as the equatorial Pacific and high latitudes of both hemispheres (Figure S1). In Sect. 3, we focus on the CO2F in the Niño3 region (5°S–5°N, 150°W–90°W) which shows notable variation of CO2F in the equatorial Pacific. This region is also the maximum variability region for SST (Gill, 1980)."

Niño4 region is the maximum variability zone for westerly wind. To show this, we have added the map for 10 m zonal and meridional wind anomalies (Figure S3), and added the following sentence in Line 460 in the revised manuscript.

"Niño4 region is the maximum variability region for the equatorial trade wind (Figure S3)."

We have added the boxes indicating Niño3 and Niño4 regions in Figure 1a and 1c in the revised manuscript.

Results 3.1

Terminology in results sections needs tightening up throughout. There are cases where increase and decrease are used when the positive and negative phase of ENSO should be referenced. More specific use of El Niño or La Niña would be helpful instead of ENSO signal.

The anomalies shown in Figures 2–5 show the ones during El Niño. Therefore, we have decided to discuss anomalies during El Niño from the climatic field in the revised manuscript. To make it clearer, we have added the following sentence in Line 356 in Section 3.1 in the revised manuscript:

"In the following, we describe the anomalies during El Niño periods, while the opposite applies during La Niña periods."

In addition, in order to describe more clearly our results, we have rewritten Sect. 3 entirely.

Line 149-156, this feels important but difficult to follow, please rephrase.

To more clearly describe the pCO2 change associated with the changes in DIC concentration and in temperature, we have rewritten Lines 149–156 in the original manuscript as follows:

"In NEW-assim, NEW, and OLD, $pCO_2$ decreases because the effect of the decrease in $pCO_2$ with decreasing DIC concentrations is larger than that of the increase in $pCO_2$ with warming (Figure 2). In OLD-assim, however, the effect of the increase in $pCO_2$ with warming is larger than that of OLD, and the decrease in $pCO_2$ with decreasing DIC concentrations is smaller than that of OLD, resulting in an increase in $pCO_2$." (Lines 357–360 in the revised manuscript)

Results 3.2
Line 179-180, this is confusing, enhanced SST anomaly during both positive and negative phases of ENSO?

In the revised manuscript, Sections 3.2 and 3.3 in the original manuscript have been combined into one and reorganized. (See reply to comments on Sect. 3.3) We have moved the description of the vertical velocity feedback in Lines 179–180 in the original manuscript to Line 594 in the revised manuscript. To make it clear that the positive value of vertical velocity feedback indicates the enhancement of upward vertical velocity during El Niño periods, we have rewritten Lines 606–608 in the revised manuscript as follows:

"The positive value of vertical velocity feedback indicates the enhancement of upward vertical velocity at the depth of the thermocline during El Niño periods, which is inconsistent with observations."

The opposite applies during La Niña periods. (Line 356 in the revised manuscript)

Line 196, why does strong heating cause upwelling? This needs better explanation.

The correction term in the governing equation of the ocean temperature leads to decrease in density during El Niño periods, causing enhancement of upward vertical velocity. In order to more clearly state this, we have rewritten the manuscript as follows:

"the correction term forces to raise the equatorial water temperature by $0.16 \times 10^{-6}\ °C\ s^{-1}$ during El Niño periods in order to realize observed temperature variations (Figure 5b)." (Lines

600–601)

"the warming due to data assimilation procedure during El Niño periods reduces density, leading to low-pressure anomalies. This results in anomalous cyclonic circulation and convergence, and thus enhancement of upward vertical velocity at the depth of the thermocline (Figure 5d)." (Lines 603–604).

Line 198, is it upwelling like in La Niña or is it upward mixing that means a smaller SST increase than would be expected for El Niño. The terminology needs to be tighter here. Please check phrasing like this throughout.

The positive value of vertical velocity feedback, discussed in Line 198 in the original manuscript, indicates the enhancement of the upward vertical velocity during El Niño periods, which in fact should not be occurring. To make it clear that the enhancement of upward vertical velocity was occurring during El Niño periods in OLD-assim, we have rewritten Lines 606–608 as follows:

"The positive value of vertical velocity feedback indicates the enhancement of upward vertical velocity at the depth of the thermocline during El Niño periods, which is inconsistent with observations."

The opposite applies during La Niña periods. (Line 357 in the revised manuscript)

We have rewritten Sect. 3 to make it clear that enhancement of the upward vertical velocity is occurring during El Niño periods in OLD-assim.

Line 200, does OLD-assim, with the temperature amplitude increase, suggest a future forecasts of increasing global temperature using OLD would not give realistic results?
The OLD-assim results here need to be discussed especially carefully with the right terms. Be sure about whether it is giving a result that is the same direction but less strong or a result that is the opposite direction i.e. La Niña like conditions during expected El Niño.

In the Niño3 region in OLD-assim, an upward CO2F anomaly is found when the SST shows the positive anomaly, which is opposite to observations (Figure 1b in the revised manuscript). This is because, in MIROC-ESM, the amplitude of the seasonal–decadal scale variations in ocean temperature in the upper layer of the eastern equatorial Pacific is too much smaller than in observations (Figure 3), so that the correction term on the governing equation of the ocean temperature in OLD-assim forces to raise the equatorial water temperature in order to realize observed temperature variations, leading to an unnatural variations in the vertical velocity. We do not think that this result in OLD-assim means that the future projection by

MIROC-ESM, where things are determined by the physics in the model, are unrealistic. In fact, the estimates of global warming by MIROC-ESM is not extremely different compared to other models. Friedlingstein et al. (2014, J. Climate, doi:10.1175/JCLI-D-12-00579.1) evaluated the twenty-first-century global surface warming defined as the 2081–2100 average relative to the 1986–2005 average under the concentration-driven RCP8.5 scenario in CMIP5 models (their Table 3). In MIROC-ESM, the global surface warming was estimated to be 4.7 ℃, which is larger than the inter-model mean, 3.7 ℃, but same with HadGEM2-ES.

Results 3.3

Section 3.2 and 3.3 would be better merged and restructured.

Thank you for your advice. We have combined Section 3.2 and 3.3 in the original manuscript and reorganized.

Please explain what makes the stratification in NEW setup better than OLD? Could it be applied to other CMIP models than are bad at reproducing ENSO?

In Lines 376–454 in the revised manuscript, we have described the two updates in model configuration in MIROC-ES2L. One is implementation of an updated plume model for cumulus convection with multiple cloud types where lateral entrainment rate varies vertically depending on the surrounding environment. The other is reduction of numerical diffusion by introducing highly-accurate tracer advection scheme in the ocean and by increasing vertical resolutions. We think these can be applied to other ESMs.

Does the different vertical depth levels/max-depth between NEW and OLD affect stratification and DIC storage in deeper water column?

In order to better represent the equatorial thermocline, the increased number of vertical layers in MIROC-ES2L has been adopted. Please see the reply above one. As a result, the stratification and DIC storage in the deeper layers may also change. However, we have to note that these may have also been changed by changing the advection scheme (Lines 381–454) and the model spinup time (Watanabe S. et al., 2011; Hajima et al., 2020).

Discussion

Key messages could be that one model in CMIP is not enough since they can be biased by misrepresented processes such as ENSO.

As you pointed out, each ESM has a model-specific bias, so that in future predictions multiple models need to be used and evaluated along with the uncertainties. To state this, we have rewritten Lines 659–662 as follows:
"There are many ESMs where the ENSO characteristics and/or the SST-CO2F relationship are inconsistent with observations. Causes of this discrepancy should be addresses in future studies through, for example, multi-model analysis, and also process-based uncertainty estimation will be further required in initialized climate and carbon predictions as well as projections by ESMs."

Line 255-260, not really needed here, we suggest to remove.

We removed the last paragraph in the original manuscript.

Figures
Figure 1, add R-value. Add the same graphs for NEW and OLD without assimilation.

We have added R-values in Figure 1 in the revised manuscript. The timeseries of SST and air–sea $CO_2$ flux in the Niño3 region simulated with NEW and OLD has been added as Figure S2.

Figure 2, x-axis label is not attractive, we suggest the authors use a colour-coded legend for the whole figure.

To make it easier to compare the magnitude of each term in Eq. (3) in each experiment, we have redesigned Figure 2.

Do you really need Figure 4, maybe a table would be better or stating the values in the text.

The results, which were presented in Figure 4 in the original manuscript, are now shown in Table 3 in the revised manuscript.

Combine Figures 5 and 7 for side by side comparison. Same for 8 and 6.

We have combined Figures 5 and 7, and Figures 6 and 8 as suggested.

Clarify the meaning of Figures 5-8. What does the colour scale mean? How should it be interpreted? Do we need to know timescale of response? Is it all calculated on monthly data?

The variation shown in Figures 2–5 in the revised manuscript represents the one during El Niño periods, and the opposite applies during La Niña periods. To make it clearer, we have added the following sentence in Line 356 in the revised manuscript:

"In the following, we describe the anomalies during El Niño periods, while the opposite applies during La Niña periods."

In the original manuscript, it was not clearly stated that we were discussing the anomalies from the climatic state. To clarify the processes occurring in our experiments, we have totally rewritten Sect. 3 in the revised manuscript. For example, we have rewritten the description on the process occurring in NEW-assim as follows:

"The maximum absolute value of the equatorial temperature analysis increment in NEW-assim is found at 10–40 m depths in the eastern equatorial Pacific, shallower than the depth of the thermocline (Figure 5a)." (Lines 479–481)

"The westerly wind anomalies in NEW-assim leads to weakening of upward vertical velocity along the equator during El Niño periods (Figure 5c)." (Lines 483–484)

"The weakening of upward vertical velocity causes lesser supply of the DIC-rich subsurface water to the surface layer, leading to the decrease in surface DIC concentration (Figure 5e)." (Lines 597–598)

The period of ENSO in NEW is longer than observations, and the data assimilation procedure can partially distort the model nature even in NEW-assim, so that we think that reproducing the observed timescale of ENSO is important along with the intensity of ENSO. We have added Table 2 showing the intensities and periods of ENSO in NEW, OLD, and the observation, respectively, and the following sentences in Lines 612–616 in the revised manuscript:

"In addition, the period of ENSO, which is defined as the peak of the power spectrum of one-year running mean NINO-SST, is 5.0 years in NEW, which is longer than 3.5 years of observations (see Table 2). Because the ENSO characteristics in NEW are not perfectly consistent with observations, model nature, namely responses of vertical velocity and DIC concentration in ENSO, are still distorted by the temperature analysis increment even in NEW-assim."

Figures 3–5 in the revised manuscript were drawn using monthly mean anomalies. To

make it clear, the corresponding parts have been rewritten. For example, Lines 366–368 in the revised manuscript have been rewritten as follows:

"A cross section of the monthly ocean temperature anomalies regressed onto monthly mean NINO3-SST anomalies along the equatorial Pacific is presented in Figure 3, together with the climatological annual mean depths of the 18, 20, and 22 °C isotherms."

**Importance of El Niño reproducibility for reconstructing historical CO₂ flux variations in the equatorial Pacific**

Michio Watanabe[1], Hiroaki Tatebe[1], Hiroshi Koyama[1], Tomohiro Hajima[1], Masahiro Watanabe[2], and
Michio Kawamiya[1]

[1]Research Institute for Global Change, Japan Agency for Marine-Earth Science and Technology (JAMSTEC), 3173-25,
Showa-machi, Kanazawa-ku, Yokohama, Kanagawa, 236-0001, Japan.
[2]Atmosphere and Ocean Research Institute, the University of Tokyo, 5-1-5, Kashiwanoha, Kashiwa, Chiba, 277-8564, Japan.

*Correspondence to:* Michio Watanabe (michiow@jamstec.go.jp)

**Abstract.** Based on a set of climate simulations utilizing two kinds of Earth System Models (ESMs) to which observed ocean hydrographic data are assimilated with an exactly same data assimilation procedure, we have clarified that successful simulation of observed historical air–sea CO₂ flux variations in the equatorial Pacific is tightly linked with the reproducibility of physical air–sea coupled processes. When an ESM with weaker amplitude of ENSO (El Niño Southern Oscillations) than observations was used for historical simulations with the ocean data assimilation, observed equatorial anticorrelated relationship between the sea surface temperature (SST) and air–sea CO₂ flux on seasonal–decadal timescales cannot be represented. The simulated CO₂ flux anomalies were upward (downward) during El Niño (La Niña) periods. The reason is that nonnegligible correction term on the governing equation of ocean temperature, which was added through the ocean data assimilation procedure, caused anomalously spurious equatorial upwelling (downwelling) during El Niño (La Niña) periods, which brought more (less) subsurface layer water rich in dissolved inorganic carbon (DIC) to the surface layer. This led to upward (downward) air–sea CO₂ flux anomalies during El Niño (La Niña) periods. On the other hand, such spurious vertical transport of DIC and resultant positively-correlated SST and air–sea CO₂ flux variations were not occurring in the historical simulations where the other ESM with rather realistic ENSO representation were used because the correction term arisen from the assimilation procedure was kept small enough not to disturb an anomalous advection-diffusion balance for the equatorial ocean temperature. Thus, the reproducibility of the tropical air–sea CO₂ flux variability with data assimilation can be significantly attributed to the reproducibility of ENSO in an ESM. Our results suggests that, when using data assimilation to initialize ESMs for carbon cycle predictions, the reproducibility of the internal climate variations in the model itself is of great importance.

**1 Introduction**

Since the industrial revolution, vast quantities of greenhouse gases (e.g., CO₂) have been released into the atmosphere through human activities such as fossil fuel use and land use change. Increased atmospheric CO₂ concentration

[revised manuscript text omitted]

下へ移動 [2]: Smith et al., 2007; Keenlyside et al., 2008; Pohlmann et al., 2009; Sugiura et al., 2009;

下へ移動 [3]: Mochizuki et al., 2010; Tatebe et al., 2012

上へ移動 [1]: 2009; Tommasi et al., 2017a, 2017b; Park et al., 2018). Li et al. (2016, 2019) studied the predictability of CO2F

下へ移動 [4]: et al., 2011).

移動 (挿入) [4]

書式を変更: フォント：ＭＳ 明朝

移動 (挿入) [3]

[revised manuscript text omitted]

上へ移動 [6]: (Ishii et al., 2005; Hirahara et al., 2014

tropical Pacific is more diffuse in OLD than in observation, which is partly arisen from numerical diffusion, especially in vertical advection (Tatebe and Hasumi, 2010), and this model bias is much alleviated in NEW. Correspondingly, so-called thermocline mode (e.g., Imada et al., 2006) becomes more effective and ENSO amplitude becomes larger in NEW. As the ENSO amplitude in NEW is larger than in OLD, the variation of the equatorial trade wind, which causes anomalous equatorial vertical velocity, is also larger in NEW.

To assess the variations of zonal wind associated with ENSO, we estimated the 10 m zonal wind anomalies over the NINO4 region (5°S–5°N, 160°E–150°W; the dotted line boxes in Figure 1a) which are regressed onto the NINO3-SST anomalies (Table 3). Niño4 region is the maximum variability region for the equatorial trade wind (Figure S3). Hereafter, the above-mentioned regression coefficient is referred to as wind feedback. The positive value of wind feedback in NEW (0.92 m s$^{-1}$ K$^{-1}$) indicates an westerly wind anomalies during El Niño, and this is consistent with that evaluated from the observational dataset, i.e., 1.02 m s$^{-1}$ K$^{-1}$. The wind feedback in OLD (0.46 m s$^{-1}$ K$^{-1}$) is about half of NEW and the observation.

Cross sections of the monthly upward water velocity and DIC concentration anomalies along the equator regressed onto NINO3-SST anomalies in NEW (OLD) are shown in Figure 4a and 4c (Figure 4b and 4d), respectively. By reproducing wind feedback that is consistent with the observation, the westerly wind anomalies during El Niño periods in NEW (Figure S3c) is comparable to that of the JRA55 reanalysis (Figure S3i), leading to weakening of upward vertical velocity of approximately 5 × 10$^{-6}$ m s$^{-1}$ (Figure 4a). This weakening of upward vertical velocity causes decrease in surface DIC in the eastern equatorial Pacific during El Niño periods (Figure 4c). In OLD, the smaller wind feedback and associated smaller westerly wind anomalies than in the JRA55 reanalysis (Figure S3g) leads to weakening of upward vertical velocity of just 10$^{-6}$ m s$^{-1}$ in the equatorial Pacific (Figure 4b). Although the ENSO signal in OLD is weaker than the observation, because of decrease in upward vertical velocity from normal years, the surface DIC concentration decreases during El Niño periods (Figure 4d). This is consistent with Dong et al. (2016), showing that OLD is able to qualitatively reproduce the negative correlation between SST and DIC concentration anomalies in the eastern equatorial Pacific (Figure S2b).

Next, we examined the correction term in temperature due to the data assimilation, i.e., temperature analysis increment, the final term on the right-hand side of Eq. (1), and the variations in vertical velocity and DIC concentration. Anomalies of monthly mean temperature analysis increments, vertical velocity, and DIC concentration along the equator regressed onto NINO3-SST anomalies are shown in Figure 5. The maximum absolute value of the equatorial temperature analysis increment in NEW-assim is found at 10–40 m depths in the eastern equatorial Pacific, shallower than the depth of the thermocline (Figure 5a). In NEW-assim, the wind feedback is 0.92 m s$^{-1}$ K$^{-1}$ (Table 3), which is of the same magnitude to that in NEW (0.92 m s$^{-1}$ K$^{-1}$), and the surface wind anomalies still shows similar pattern to that of the NEW (Figure S3a–d). The westerly wind anomalies in NEW-assim leads to weakening of upward vertical velocity along the equator during El Niño periods (Figure 5c). To assess the variation in equatorial vertical velocity associated with ENSO, we estimated the anomalies of the vertical velocity at the depth of the 20 °C isotherm (the depth of the thermocline) in the Niño3 region which are regressed onto the NINO3-SST anomalies. Hereafter, the regression coefficient is referred to as vertical velocity

We investigated the correction in temperature due to the data assimilation (temperature increment, the final term on the right-hand side of Eq. (1)) and the fluctuations in vertical velocity and DIC concentration in OLD-assim. The monthly mean temperature increment, vertical velocity, and DIC concentration along the Equator regressed onto NINO3-SST …

上へ移動 [5]: In Sect.

feedback. The vertical velocity feedback in NEW-assim is estimated to be $-4.5 \times 10^{-7}$ m s$^{-1}$ K$^{-1}$, which is not significantly different from that in NEW ($-3.9 \times 10^{-7}$ m s$^{-1}$ K$^{-1}$) (Table 3). The negative value of vertical velocity feedback in NEW-assim indicates the weakening of upward vertical velocity at the depth of the thermocline during El Niño periods in the eastern equatorial Pacific (Figure 5c). The weakening of upward vertical velocity causes lesser supply of the DIC-rich subsurface water to the surface layer, leading to the decrease in surface DIC concentration (Figure 5e). In OLD, the temperature variations associated with ENSO at the depth of the thermocline in the eastern equatorial Pacific is smaller than observed (see Figure 3b and 3c), so that the correction term forces to raise the equatorial water temperature by $0.16 \times 10^{-6}$ °C s$^{-1}$ during El Niño periods in order to realize observed temperature variations (Figure 5b). The wind feedback in OLD-assim is $0.48$ m s$^{-1}$ K$^{-1}$ (Table 3), which is the same as in OLD, and the map of the wind speed anomalies shows a similar pattern to that of the OLD (Figure S3e–h); however, the warming due to data assimilation procedure during El Niño periods reduces density, leading to low-pressure anomalies. This results in anomalous cyclonic circulation and convergence, and thus enhancement of upward vertical velocity at the depth of the thermocline (Figure 5d). The vertical velocity feedback in OLD-assim is $4.1 \times 10^{-7}$ m s$^{-1}$ K$^{-1}$, which has an opposite sign to OLD, $-4.9 \times 10^{-7}$ m s$^{-1}$ K$^{-1}$ (Table 3). The positive value of vertical velocity feedback indicates the enhancement of upward vertical velocity at the depth of the thermocline during El Niño periods, which is inconsistent with observations. This spurious enhancement of upward vertical velocity during El Niño periods causes the increase in the surface DIC concentration (Figure 5f), leading to positive correlation between SST and CO2F (Figure 1d), contrary to observations. We have to note here that even in the NEW-assim, the vertical velocity distribution (Figure 5c) is still different from NEW (Figure 4a) because of the temperature analysis increment. As already discussed, the intensity of ENSO in NEW is slightly stronger than observed (Table 2). In addition, the period of ENSO, which is defined as the peak of the power spectrum of one-year running mean NINO-SST, is 5.0 years in NEW, which is longer than 3.5 years of observations (see Table 2). Because the ENSO characteristics in NEW are not perfectly consistent with observations, model nature, namely responses of vertical velocity and DIC concentration in ENSO, are still distorted by the temperature analysis increment even in NEW-assim. This indicates that further model improvements are needed.

**4 Discussion and Summary**

In the present study, comparing the results of two ESMs to which observed ocean hydrographic data are assimilated, we have clarified that representation of the processes in the equatorial climate system is important to reproduce the observed anticorrelated relationship between SST and CO2F in the equatorial Pacific. In the case where the ocean temperature and salinity observations were assimilated into an ESM with weaker amplitude of ENSO than observations, the correction term on the governing equation of the ocean temperature, which was introduced in the data assimilation procedure, caused spurious upwelling (downwelling) anomalies along the equator during El Niño (La Niña) periods, leading to more (less) supply of the DIC-rich subsurface water to the surface layer. Due to the resultant increase (decrease) of the surface

DIC concentration, the upward (downward) CO2F anomalies during El Niño (La Niña) periods was induced, which was inconsistent with observation. In the case where the ocean temperature and salinity observations were assimilated into the other ESM with rather realistic ENSO representation, anticorrelated relationship between SST and CO2F was reproduced.

Focusing on the CO2F fluctuations associated with ENSO in the equatorial Pacific, Dong et al. (2016) analyzed the results of the CMIP5 ESMs. They showed that only a portion of CMIP5 ESMs (including MIROC-ESM) could reproduce the observed anticorrelated relationship between SST and CO2F. Bellenger et al. (2014) evaluated the reproducibility of ENSO in the CMIP5 models. They reported that most CMIP5 climate models and ESMs underestimate the amplitude of the wind stress feedback by 20%–50%, and that only 20% of CMIP5 models have relative error within 25% of the observed value. There are many ESMs where the ENSO characteristics and/or the SST-CO2F relationship are inconsistent with observations. Causes of this discrepancy should be addresses in future studies through, for example, multi-model analysis, and also process-based uncertainty estimation will be further required in initialized climate and carbon predictions as well as projections by ESMs.

**Data availability**

The model outputs of MIROC-ES2L (Hajima et al., 2019) are available through the Earth System Grid Federation (ESGF) (https://doi.org/10.22033/ESGF/CMIP6.5602). The model outputs of MIROC-ESM is also available at ESGF, https://esgf-node.llnl.gov/projects/cmip5/. The CMIP6 forcing data is version 6.2.1, and the CMIP5 forcing data is described at https://pcmdi.llnl.gov/mips/cmip5/forcing.html. The JRA55 reanalysis wind dataset is available at https://jra.kishou.go.jp/JRA-55/index_en.html. The COBE-SST2 dataset is available at https://www.esrl.noaa.gov/psd/data/gridded/data.cobe2.html. The postprocessing scripts used for this research and the data used in the figures can be obtained online (https://osf.io/mpk52).

**Author contribution**

MiW, HT, MaW, and MK contributed to the experiment design. MiW and HK embedded the ocean data assimilation system into the ESMs. MiW and TH performed the experimental simulations. MiW analyzed the model output and drafted the paper. All authors discussed the results, and commented on and edited the manuscript.

**Competing interests**

[revised manuscript text omitted]

↵ ... [17]

[Figure]

[Figure]

(a) W reg. Nino3, NEW  (b) W reg. Nino3, OLD

W [10⁻⁶ m s⁻¹]

(c) DIC reg. Nino3, NEW  (d) DIC reg. Nino3, OLD

DIC [μmol L⁻¹]

985   **Figure 4. Anomalies of equatorial vertical velocity (upper panels) and DIC (lower panels) regressed onto NINO3-SST anomalies for NEW (left) and OLD (right). Contour intervals are 0.5 × 10⁻⁶ m s⁻¹ in (a,b) and 2 μmol L⁻¹ in (c,d), respectively. Thick solid lines indicates the climatological-mean isotherms of the 18, 20, and 22 °C.**

[Figure]

(a) W reg. Nino3 Index

W [10⁻⁶ m s⁻¹]

(b) DIC reg. Nino3 Index

DIC [μmol L⁻¹]

Figure 5. Cross sections along the Equator of monthly (a) upward velocity upper panels)

書式を変更: 上付き

書式を変更: 上付き

書式を変更: 上付き

[Figure]

[Figure]

[Figure]

**Figure 5. Equatorial temperature analysis increments (top panels), vertical velocity anomalies (middle panels) and DIC anomalies (bottom panels) regressed onto NINO3-SST anomalies for NEW-assim (left) and OLD-assim (right). Contour intervals are 0.02 × 10⁻⁶ °C s⁻¹ in (a,b), 0.5 × 10⁻⁶ m s⁻¹ in (c,d) and 2 μmol L⁻¹ in (e,f), respectively. Thick solid lines indicates the climatological-mean isotherms of the 18, 20, and 22 °C.**

1010

**Table 1. Correlation coefficients between detrended one-year running mean NINO3-SST and NINO3-CO2F anomalies in NEW-assim, NEW, OLD-assim, OLD, and observations. The correlations coefficients in NEW-assim, NEW, OLD-assim, and OLD are for the period from 1961 to 2005 (Figures 1 and S2), and that in observations are for the period from 1982 to 2005.**

|  | NEW-assim | NEW | OLD-assim | OLD | Observation |
|---|---|---|---|---|---|
| Corr. Coeff. | −0.50 | −0.85 | 0.44 | −0.67 | −0.75 |

**Table 2. The intensity and period of ENSO in NEW, OLD, and observations calculated from the one-year running mean NINO3-SST anomalies for the period from 1961 to 2005.**

|  | NEW | OLD | Observation |
|---|---|---|---|
| Intensity of ENSO [°C] | 1.17 | 0.43 | 0.80 |
| Period of ENSO [yr] | 5.0 | 4.5 | 3.5 |

**Table 3. The wind feedback computed as the monthly 10 m zonal wind anomalies in the Niño4 region which is regressed onto the monthly NINO3-SST anomalies and the vertical velocity feedback computed as the monthly vertical velocity anomalies at the depth of the 20 °C isotherm in the Niño3 region which is regressed onto the monthly NINO3-SST anomalies in NEW-assim, NEW, OLD, and OLD-assim. The wind feedback is also evaluated from the observation dataset.**

|  | NEW-assim | NEW | OLD-assim | OLD | Observation |
|---|---|---|---|---|---|
| Wind feedback [m s⁻¹ K⁻¹] | 0.92 | 0.92 | 0.48 | 0.46 | 1.02 |
| Vertical velocity feedback [m s⁻¹ K⁻¹] | $-4.5\times10^{-7}$ | $-3.9\times10^{-7}$ | $4.1\times10^{-7}$ | $-4.9\times10^{-7}$ | N/A |

[Figure]

**Figure 7. As Figure 5 but for NEW.**

**Figure 8. As Figure 6 but for NEW-assim.**

---

## Author Response (AR2)

Dear Dr. Hoppema,

Thank you very much for handling our manuscript and for taking the time to read it. Following the comments of the editor and reviewers, we have revised the manuscript. Below, we would like to describe how we have revised our manuscript point by point. Line numbers below refer to the revised, tracked-changes version manuscript. The comments of the editor and reviewers are indicated in black text and our responses are indicated in red text.

**Reply to the editor's comment:**

L41-42 "CO2F is anomalously downward during El Niño, and vice-vasa during La Niña" I am not sure what you mean here. Is the flux direction different or is the flux not as high as previously? Please change wording to make this clear.

CO2F anomaly is downward during El Niño. In order to clarify it, we have rewritten the sentence as "CO2F anomaly is downward during El Niño, and vice-versa during La Niña." (Lines 53–54 in the revised manuscript)

L42 typo: vice-versa

**Corrected. (Line 54 in the revised manuscript)**

L47 Please add some more words on what the Paris Agreement is, i.e. what conference etc. At the moment most people are familiar with the Paris Agreement, but this might be different in a few years from now.

In the revised manuscript, to describe the framework of the Paris Agreement, we have added the following sentence in Line 59 in the revised manuscript: "The Paris Agreement is an agreement within the United Nations Framework Convention on Climate Change (UNFCCC, 2015) providing the framework of measures from 2021 to 2030 to act against climate change."

Data availability: Please check if data providers wish particular acknowledgements for data use.

We have checked the web site of the data providers. We realized that we were not describing the availability of the SOM-FFN dataset, so that we have added that to the "Data

availability" section. In addition, following the request by the data provider, we have cited Landschützer et al. (2017) in Line 165 in the revised manuscript.

L343-346 Please update the reference to the final article in Geosci. Model Dev.

We have updated the reference.

L376-377 Any update of this submission from 2019?

Kawamiya et al. was recently accepted and we have updated the reference.

L393 CO2 (subscript)

**Corrected.**

Table 3 title: The first sentence is hard to understand. Please make clear, for example by splitting into two sentences.

We agree that the first sentence of the caption of Table 3 in the second manuscript was too long and difficult to understand. In the revised manuscript, we have rewritten it as follows: "The wind feedback and the vertical velocity feedback in NEW-assim, NEW, OLD, and OLD-assim. The wind feedback is computed as the monthly 10 m zonal wind anomalies in the Niño4 region regressed onto the monthly NINO3-SST anomalies, and the vertical velocity feedback the monthly vertical velocity anomalies at the depth of the 20 °C isotherm in the Niño3 region regressed onto the monthly NINO3-SST anomalies. The wind feedback is also evaluated from the observation dataset."

**Reply to referee #1:**

Thank you very much for invaluable comments on our manuscript. According to the comment, we have revised the manuscript as follows. We hope that the revised manuscript is now suitable for publication in the journal.

**Reply to comments:**

(Referee #1) "I am basically satisfied with the response of the authors, and would like to recommend it for the publication in OS. Nevertheless, there is an expression that should be verified before its publication. In the abstract, the sentences including "seasonal-decadal timescales" probably need to be modified. As I understand, most of the analyses are based on the one-year running mean filtered data, in which the seasonal signs have been removed. I have not seen any discussion of results on the seasonal timescale."

As the referee #1 pointed out, we do not discuss variations in seasonal timescale in this study. We have changed the word "seasonal" to "interannual" in the revised manuscript. (Lines 15 and 80)

**Reply to referee #3:**

Thank you very much for invaluable comments on our manuscript. According to the comments, we have revised the manuscript as follows. We hope that the revised manuscript will be more suitable for publication in the journal.

**Reply to comments:**

(Referee #3) "The revised manuscript is largely improved. The results and figures are well organised and presented in the revised manuscript. I would suggest for publication after clarifying the points below.

**Thank you for your comments.**

Abstract line 12: "observed historical" -> "observed"

**Corrected.**

Abstract line 20-24: the sentence is unclear and needs to be rephrased.

The sentence in Lines 20–24 in the second manuscript was too long and the points we wanted to make were not clear. In the revised manuscript, we have rewritten it as follows: "On the other hand, in the historical simulations where the observational data were assimilated into the other ESM with more realistic ENSO representation, the correction term associated with the assimilation procedure was kept small enough so as not to disturb an anomalous

advection-diffusion balance for the equatorial ocean temperature. Consequently, spurious vertical transport of DIC and resultant positively-correlated SST and air–sea CO2 flux variations did not occur." (Line 20–24 in the revised manuscript)

The 2 sentences in line 16 and line 20 in abstract are repetitive, i.e., "The simulated CO2 flux anomalies were upward (downward) during El Nino (La Nina) periods. " and "The simulated CO2 flux anomalies were upward (downward) during El Nino (La Nina) periods." Suggest to leave one and specify the region: "The simulated CO2 flux anomalies were upward (downward) during El Nino (La Nina) periods in the equatorial Pacific."

Thank you for your suggestion. We have removed the sentence "The simulated CO2 flux anomalies were upward (downward) during El Nino (La Nina) periods in the equatorial Pacific" in Line 20 in the second manuscript, and added the phrase "in the equatorial Pacific" in Line 16 in the revised manuscript.

Lines 98-101, "The analysis increment is calculated from  $\Delta Xa = Xa(0) - X(0)$ , where Xa(0) is the analysis and X(0) is the model first guess at t = 0; this term is held constant during the analysis interval. For Xa(0) and X(0), we used anomalies from monthly mean climatology during 1961–2000 in observations and models, respectively." This method description is unclear. Does the IAU method always use the difference between analysis and model first guess at the initial step, i.e., t=0? How long is the analysis interval? Are the anomalies relative to climatology of model or observations?

During the analysis interval, the difference between analysis and model first guess at the initial step, i.e., t=0, is used. We have rewritten the phrase in Lines 99–100 in the second manuscript as "this term is kept unchanged during the analysis interval from t = 0 to t =  $\tau$ ." (Lines 119–120 in the revised manuscript). The analysis time interval  $\tau$  in this study is 1 day (Line 117 in the revised manuscript). For  $X^{a}(0)$ , we used observed anomalies with respect to observed monthly mean climatology during 1961–2000. For X(0), simulated anomalies in NEW-assim (OLD-assim) with respect to monthly mean climatology in NEW (OLD) were used. We have rewritten the sentence in Line 100–101 in the second manuscript as follows: "For  $X^{a}(0)$ , we used observed anomalies with respect to observed monthly mean climatology during 1961–2000. For X(0), simulated anomalies in NEW-assim (OLD-assim) with respect to observed monthly mean climatology during 1961–2000. For  $X^{a}(0)$ , we used observed anomalies with respect to monthly mean climatology during 1961–2000. For X(0), simulated anomalies in NEW-assim (OLD-assim) with respect to observe monthly mean climatology during 1961–2000. For X(0), simulated anomalies in NEW-assim (OLD-assim) with respect to observe monthly mean climatology during 1961–2000. For X(0), simulated anomalies in NEW-assim (OLD-assim) with respect to observe monthly mean climatology during 1961–2000. For X(0), simulated anomalies in NEW-assim (OLD-assim) with respect to monthly mean climatology during 1961–2000. For X(0), simulated anomalies in NEW-assim (OLD-assim) with respect to monthly mean climatology during 1961–2000. For X(0), simulated anomalies in NEW-assim (OLD-assim) with respect to monthly mean climatology in NEW (OLD) were used." (Lines 120–122 in the revised manuscript)

The figures are largely improved, the new presentation of different component contribution to  $\Delta pCO2$  in Fig.2 is compact and informative. I have a question to the calculation as stated in Lines 163-164: "...we estimated  $\Delta X$  by averaging monthly mean X anomalies regressed on the NINO3-SST anomalies over the entire Nino3 region,..." Why does  $\Delta X$  involve regression to NINO3-SST anomalies? This part is not stated in equation 2.

In order to estimate pCO2 variations due to ENSO-related X variations, we calculated monthly mean X anomalies regressed on the NINO3-SST anomalies (X = T, S, DIC, or Alk). To clarify this, we have rewritten Lines 162–165 in the second manuscript as follows: " $\partial pCO_2/\partial X$  in C(X) term in Eq. (3) (X=T, S, DIC, or Alk) was estimated based on the climatological annual mean T, S, DIC, and Alk at the sea surface within the Niño3 region in each experiment.  $\Delta X$  in C(X) ( $\Delta pCO_2$  on the left-hand side of Eq. (3)) is the variation of X ( $pCO_2$ ) associated with ENSO and was calculated by averaging monthly mean X ( $pCO_2$ ) anomalies regressed on NINO3-SST anomalies over the entire Niño3 region. Note that the NINO3-SST anomalies are standardized by standard deviation." (Lines 192–196 in the revised manuscript)

As the regression of temperature increment to NINO3-SST shown in Fig. 5 a-b is not linearly additive to the regression of temperature in Fig. 3 to get the temperature regression of the assimilation run. I am curious how the equatorial ocean temperature in assimilation runs regressed to NINO3-SST look like. Maybe the authors could add a figure of results from the assimilation runs as Fig. 3 in supplementary.

Thank you for your suggestion. We have added Figure S3 in supplementary, which shows the anomalies of the equatorial ocean temperature regressed onto NINO3-SST anomalies for assimilation runs. The variation of water temperature in OLD-assim is slightly weaker than the observations.

**Importance of El Niño reproducibility for reconstructing historical CO2 flux variations in the equatorial Pacific**

Michio Watanabe1, Hiroaki Tatebe1, Hiroshi Koyama1, Tomohiro Hajima1, Masahiro Watanabe2, and 5 Michio Kawamiya1

1Research Institute for Global Change, Japan Agency for Marine-Earth Science and Technology (JAMSTEC), 3173-25, Showa-machi, Kanazawa-ku, Yokohama, Kanagawa, 236-0001, Japan.

2Atmosphere and Ocean Research Institute, the University of Tokyo, 5-1-5, Kashiwanoha, Kashiwa, Chiba, 277-8564, Japan. *Correspondence to:* Michio Watanabe (michiow@jamstec.go.jp)

- 10 **Abstract.** Based on a set of climate simulations utilizing two kinds of Earth System Models (ESMs) to which observed ocean hydrographic data are assimilated with an exactly same data assimilation procedure, we have clarified that successful simulation of observed air-sea CO2 flux variations in the equatorial Pacific is tightly linked with the reproducibility of physical air-sea coupled processes. When an ESM with weaker amplitude of ENSO (El Niño Southern Oscillations) than observations was used for historical simulations with the ocean data assimilation, observed equatorial anticorrelated
- 15 relationship between the sea surface temperature (SST) and air-sea CO2 flux on interannual-to-decadal timescales cannot be represented. The simulated CO2 flux anomalies were upward (downward) during El Niño (La Niña) periods in the equatorial Pacific. The reason is that nonnegligible correction term on the governing equation of ocean temperature, which was added through the ocean data assimilation procedure, caused anomalously spurious equatorial upwelling (downwelling) during El Niño (La Niña) periods, which brought more (less) subsurface layer water rich in dissolved inorganic carbon (DIC) to the
- 20 surface layer. On the other hand, in the historical simulations where the observational data were assimilated into the other ESM with more realistic ENSO representation, the correction term associated with the assimilation procedure was kept small enough so as not to disturb an anomalous advection-diffusion balance for the equatorial ocean temperature. Consequently, spurious vertical transport of DIC and resultant positively-correlated SST and air-sea CO2 flux variations did not occur. Thus, the reproducibility of the tropical air-sea CO2 flux variability with data assimilation can be significantly attributed to 25 the reproducibility of ENSO in an ESM. Our results suggest that, when using data assimilation to initialize ESMs for carbon
- cycle predictions, the reproducibility of the internal climate variations in the model itself is of great importance.

**1** Introduction**

Since the industrial revolution, vast quantities of greenhouse gases (e.g., CO2) have been released into the atmosphere through human activities such as fossil fuel use and land use change. Increased atmospheric CO2 concentration leads to global warming, while both the oceanic and the terrestrial ecosystems absorb atmospheric CO2 and are considered to

1

| 削隊                                                                                                                                                                                                                                                                                                                                                                             | t: historical                                                                                                                                                                                                                                                                                               |
|--------------------------------------------------------------------------------------------------------------------------------------------------------------------------------------------------------------------------------------------------------------------------------------------------------------------------------------------------------------------------------|-------------------------------------------------------------------------------------------------------------------------------------------------------------------------------------------------------------------------------------------------------------------------------------------------------------|
|                                                                                                                                                                                                                                                                                                                                                                                |                                                                                                                                                                                                                                                                                                             |
|                                                                                                                                                                                                                                                                                                                                                                                |                                                                                                                                                                                                                                                                                                             |
| 削除                                                                                                                                                                                                                                                                                                                                                                             | seasonal-                                                                                                                                                                                                                                                                                                   |
| 削服                                                                                                                                                                                                                                                                                                                                                                             |                                                                                                                                                                                                                                                                                                      |
| HUR                                                                                                                                                                                                                                                                                                                                                                            | Se .                                                                                                                                                                                                                                                                                                        |
|                                                                                                                                                                                                                                                                                                                                                                                |                                                                                                                                                                                                                                                                                                             |
|                                                                                                                                                                                                                                                                                                                                                                                |                                                                                                                                                                                                                                                                                                             |
|                                                                                                                                                                                                                                                                                                                                                                                |                                                                                                                                                                                                                                                                                                             |
|                                                                                                                                                                                                                                                                                                                                                                                |                                                                                                                                                                                                                                                                                                             |
|                                                                                                                                                                                                                                                                                                                                                                                |                                                                                                                                                                                                                                                                                                             |
|                                                                                                                                                                                                                                                                                                                                                                                |                                                                                                                                                                                                                                                                                                             |
| Bil Rd                                                                                                                                                                                                                                                                                                                                                                         | This lad to unward (downward) air san CO: flux anomali                                                                                                                                                                                                                                                      |
| 削険
duri                                                                                                                                                                                                                                                                                                                                                                     | ♣ This led to upward (downward) air-sea CO₂ flux anomalic
no El Niño (1 a Niña) periods                                                                                                                                                                                                                  |
| 削険
duri                                                                                                                                                                                                                                                                                                                                                                     | ♣ This led to upward (downward) air-sea CO 2 flux anomaliong El Niño (La Niña) periods                                                                                                                                                                                                           |
| 削援
duri                                                                                                                                                                                                                                                                                                                                                                     | ♥ This led to upward (downward) air–sea CO₂ flux anomalio
ng El Niño (La Niña) periods                                                                                                                                                                                                                   |
| 削版
duri
削版                                                                                                                                                                                                                                                                                                                                                               | ই: This led to upward (downward) air-sea CO2 flux anomalie
ng El Niño (La Niña) periods
ই: such                                                                                                                                                                                                       |
| 削版
duri
削版
削版                                                                                                                                                                                                                                                                                                                                                         | This led to upward (downward) air-sea CO 2 flux anomalie
ng El Niño (La Niña) periods      such     were                                                                                                                                                                                      |
| 削隊
duri
削隊                                                                                                                                                                                                                                                                                                                                                               | This led to upward (downward) air-sea CO 2 flux anomalic
ng El Niño (La Niña) periods     such     were                                                                                                                                                                                       |
| 削
duri
削
関
削
関                                                                                                                                                                                                                                                                                                                                                  |                                                                                                                                                                                                                                                                                                             |
| 削援
duri
削援
削援
削援
rath                                                                                                                                                                                                                                                                                                                                           |                                                                                                                                                                                                                                                                                                             |
| 削援
duri
削援
削援
rath
corr                                                                                                                                                                                                                                                                                                                                         | 
[revised manuscript text omitted]